# Sequential co-reduction of nitrate and carbon dioxide enables selective urea electrosynthesis

Yang Li[1,2,6], Shisheng Zheng[1,6], Hao Liu[1], Qi Xiong[1], Haocong Yi[1], Haibin Yang [1], Zongwei Mei[1], Qinghe Zhao[1], Zu-Wei Yin[1], Ming Huang [3] ✉, Yuan Lin[4], Weihong Lai[5], Shi-Xue Dou [5], Feng Pan [1] & Shunning Li [1] ✉

Despite the recent achievements in urea electrosynthesis from co-reduction of nitrogen wastes (such as $NO_3^-$) and $CO_2$, the product selectivity remains fairly mediocre due to the competing nature of the two parallel reduction reactions. Here we report a catalyst design that affords high selectivity to urea by sequentially reducing $NO_3^-$ and $CO_2$ at a dynamic catalytic centre, which not only alleviates the competition issue but also facilitates C−N coupling. We exemplify this strategy on a nitrogen-doped carbon catalyst, where a spontaneous switch between $NO_3^-$ and $CO_2$ reduction paths is enabled by reversible hydrogenation on the nitrogen functional groups. A high urea yield rate of 596.1 μg mg$^{-1}$ h$^{-1}$ with a promising Faradaic efficiency of 62% is obtained. These findings, rationalized by in situ spectroscopic techniques and theoretical calculations, are rooted in the proton-involved dynamic catalyst evolution that mitigates overwhelming reduction of reactants and thereby minimizes the formation of side products.

The recent surge in literature devoted to electrochemical synthesis of urea has been fuelled by the desperate need for both energy conservation and $CO_2$ fixation to mitigate climate change. As one of the most frequently used nitrogen fertilizers[1,2], urea can be synthesized at the industrial level via the consecutive reactions of $N_2 + H_2 \rightarrow NH_3$ and $NH_3 + CO_2 \rightarrow$ urea, both of which are energy-intensive and require harsh conditions[3–8]. Renewable electricity-driven production of urea using nitrogen species (e.g., $N_2$, nitrate, nitrite, NO) and $CO_2$ as feedstocks[9–21], offers a promising alternative to the conventional route (Supplementary Fig. 1), but there is still a lack of studies that embody practical solutions to urea electrosynthesis at large scale[22,23]. The most daunting challenge is to discover a selective catalyst for efficient C−N coupling after moderate hydrogenation of the reactants, while inhibiting their conversion into side products such as $NH_3$ and CO. In regard to hydrogenation and

C−N coupling process, the fixed nitrogen is more appealing than $N_2$ as the nitrogen source, since $N_2$ molecule has an exceedingly high dissociation energy for the triple bond (941 kJ mol$^{-1}$) and an inferior solubility in water ($6.24 \times 10^{-4}$ mol L$^{-1}$ atm$^{-1}$)[24–26]. Although direct reaction between adsorbed $N_2$ and CO was proposed, kinetic restrictions would prevent the attainment of a satisfactory yield rate of urea. In contrast, electrochemical co-reduction of nitrate ($NO_3^-$) and $CO_2$ is much easier to realize[10,13,14], which, when steered towards urea formation, has the tantalizing prospect of industrialization given the easy access of $NO_3^-$ from industrial wastewater and domestic sewage (Supplementary Fig. 2)[27–29]. The problem, however, is that the hydrogenation processes of $NO_3^-$ and $CO_2$ actually compete with each other, not to mention that both processes should take place at virtually the same site to permit facile C−N coupling[20,21,30]. Indeed, with few exceptions, side reactions always predominate over

[1]School of Advanced Materials, Peking University, Shenzhen Graduate School, Shenzhen, Guangdong 518055, China. [2]Hydrogen Energy Institute, Zhejiang University, Hangzhou, Zhejiang 310027, China. [3]Institute of Fundamental and Frontier Sciences, University of Electronic Science and Technology of China, Chengdu 611731, China. [4]Institute of Chemistry, Chinese Academy of Sciences, Beijing 100190, China. [5]Institute for Superconducting and Electronic Materials, University of Wollongong, Wollongong, NSW 2522, Australia. [6]These authors contributed equally: Yang Li, Shisheng Zheng. ✉e-mail: huangming@uestc.edu.cn; lisn@pku.edu.cn

urea formation on a variety of electrocatalysts due to the over-whelming reduction of one reactant over the other[25,31–35].

To alleviate the competition between concurrent $NO_3^-$ and $CO_2$ reduction, time-staggering of both reactions could serve as a viable strategy; that is, the reduction of each reactant is spatially coincident but temporally separated and favoured at different stages. Figure 1a depicts the reaction timeline of the conventional mechanism in the reduction process for urea synthesis, with $NO_3^-$ expected to be relatively more reactive than $CO_2$ on the catalyst. In comparison, the sequential mechanism (Fig. 1b) avoids the predominance of nitrate reduction reaction (NtrRR) along the whole reaction path. In the beginning, NtrRR is assumed to take place prior to $CO_2$ reduction reaction (CO2RR), and at this stage, CO2RR is suspended due to inferior kinetics. At the point where NtrRR process encounters a major reaction barrier, the further reduction of the corresponding intermediate is halted, with CO2RR simultaneously switched on. Before the release of CO2RR products, C−N coupling utilizing the preformed NtrRR intermediates nearby should take place and direct the reaction to urea formation. As a prerequisite for this sequential reduction process, the catalyst should be capable of switching its catalytic activity in favour of CO2RR after the initiation of NtrRR. However, we know of no previous attempts to explore such catalyst. Thanks to recent studies reporting the Faradaic pseudocapacitance behaviour of nitrogen-doped carbon (NC) materials[36,37], we were intrigued by the fact that the nitrogen species in $sp^2$-hybridized carbon could undergo reversible hydrogenation, implying their possible use as dynamic active sites during catalysis. The feasibility of reversible formation and cleavage of N−H bonds on NC could provide an attractive platform for self-tuning of the catalytic activity[38,39], and we speculate that this feature could regulate the formation sequence of intermediates during co-reduction reactions.

In this work, we compare the electrochemical performance between a defective NC catalyst and a $Cu_1$/NC single-atom catalyst that is prepared using the same procedure as the former except for the addition of a Cu precursor. Previous studies have revealed that the activity of pyridinic/pyrrolic nitrogen-coordinated single-atom catalysts is mainly derived from the isolated metal centres[40,41]. In particular, single-atom Cu species were perceived to be active for catalysing NtrRR[25] and CO2RR[42]. Here we demonstrate that while $Cu_1$/NC excels in NtrRR along with decent activity for CO2RR, the strong competition between both reduction processes throughout the whole reaction path results in low production of urea. In contrast, the NC catalyst not only triggers sequential reduction of $NO_3^-$ and $CO_2$, but also enables facile C−N coupling, which confers extraordinary catalytic performance for urea electrosynthesis. Benefiting from these features, a urea yield rate of 596.1 μg mg⁻¹ h⁻¹ with a Faradaic efficiency (FE) of 62% is achieved on NC at −0.5 V versus reversible hydrogen electrode (RHE),

which is superior to most of the previously reported catalysts. This sequential reduction behaviour stems from the reaction-driven evolution of the NC catalyst, shaping a seesaw scenario: the seesaw of the reaction is initially tilted to NtrRR in the presence of C=N−H species on the catalyst, during which the N−H bonds are cleaved and the catalytic centres become activated for CO2RR, thus tipping the seesaw over and turning the reaction to *CO formation and further C−N coupling. After the whole reaction process, the catalytic centres would be spontaneously restored to the initial state. This dynamic reversibility can endow a high propensity for urea formation and hence gives rise to the unprecedented inhibition of side reactions, which offers a strategy to design highly selective catalysts for urea electrosynthesis.

## Results

### Structural characterization

The NC was prepared by a simple one-pot pyrolysis method using glucose and dicyandiamide as carbon and nitrogen sources, respectively[9,43–45], while $Cu_1$/NC was obtained by further addition of $Cu(NO_3)_2$ (Supplementary Fig. 3). X-ray diffraction (XRD) result of NC shows a broad diffraction peak at 21.1° (Supplementary Fig. 4), which can be attributed to the typical graphitic (002) plane. An identical pattern is displayed for $Cu_1$/NC, implying the absence of metal nanoparticles. No peak can be assigned to the in-plane long-range order structure of $C_3N_4$ for both samples. Electron microscopy results substantiate the sheet-like morphology of both samples and reveal an abundance of folds and wrinkles (Supplementary Figs. 5–7). Figure 2a, d presents the aberration-corrected high-angle annular dark-field scanning transmission electron microscopy (HAADF-STEM) images of NC and $Cu_1$/NC, with the latter exhibiting bright spots that correspond to single Cu atoms dispersed across the substrate. Energy dispersive X-ray spectroscopy (EDS) elemental mapping indicates a homogenous distribution of nitrogen in the NC sample (Supplementary Fig. 6). This homogenous distribution is well preserved in $Cu_1$/NC (Supplementary Fig. 7), where the nitrogen species could serve as effective sites for capturing and dispersing the Cu atoms. The atomic dispersion of Cu is confirmed by extended X-ray absorption fine structure (EXAFS) spectroscopy, showing the presence of a strong peak assigned to the Cu-N bond (Supplementary Fig. 8). Fourier-transform infrared (FTIR) spectroscopy further indicates the formation of N−H species in the π-conjugated network of graphitic carbon (Supplementary Fig. 9)[46]. Electrons donated by the N atoms can be directly probed by the electron paramagnetic resonance (EPR) spectra (Supplementary Fig. 10). Moreover, the co-appearance of $sp^2$ and $sp^3$ peaks in the Raman spectrum of NC (Supplementary Fig. 11) indicates the existence of numerous defects in the architecture, which are inherent to the calcination process during synthesis. Related to this feature is a high specific surface area of 879.59 m² g⁻¹ as estimated from the

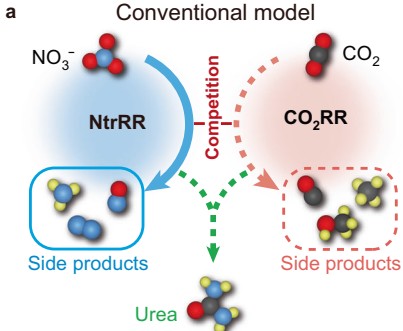

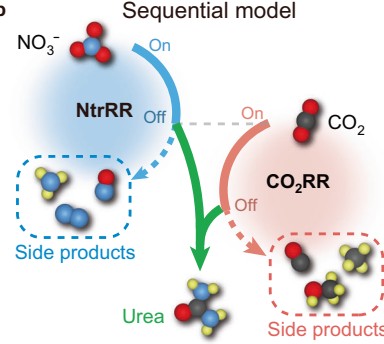

**Fig. 1 | Conventional and sequential models for urea electrosynthesis.**
**a** Concurrent occurrence of NtrRR and CO2RR, leading to inferior production of urea and pronounced formation of side products, such as those from NtrRR.

**b** Sequential combination of NtrRR and CO2RR, successively biasing the competition between both reactions in favour of one and resulting in high selectivity for urea synthesis. Colour codes: N, blue; O, red; C, black; H, yellow.

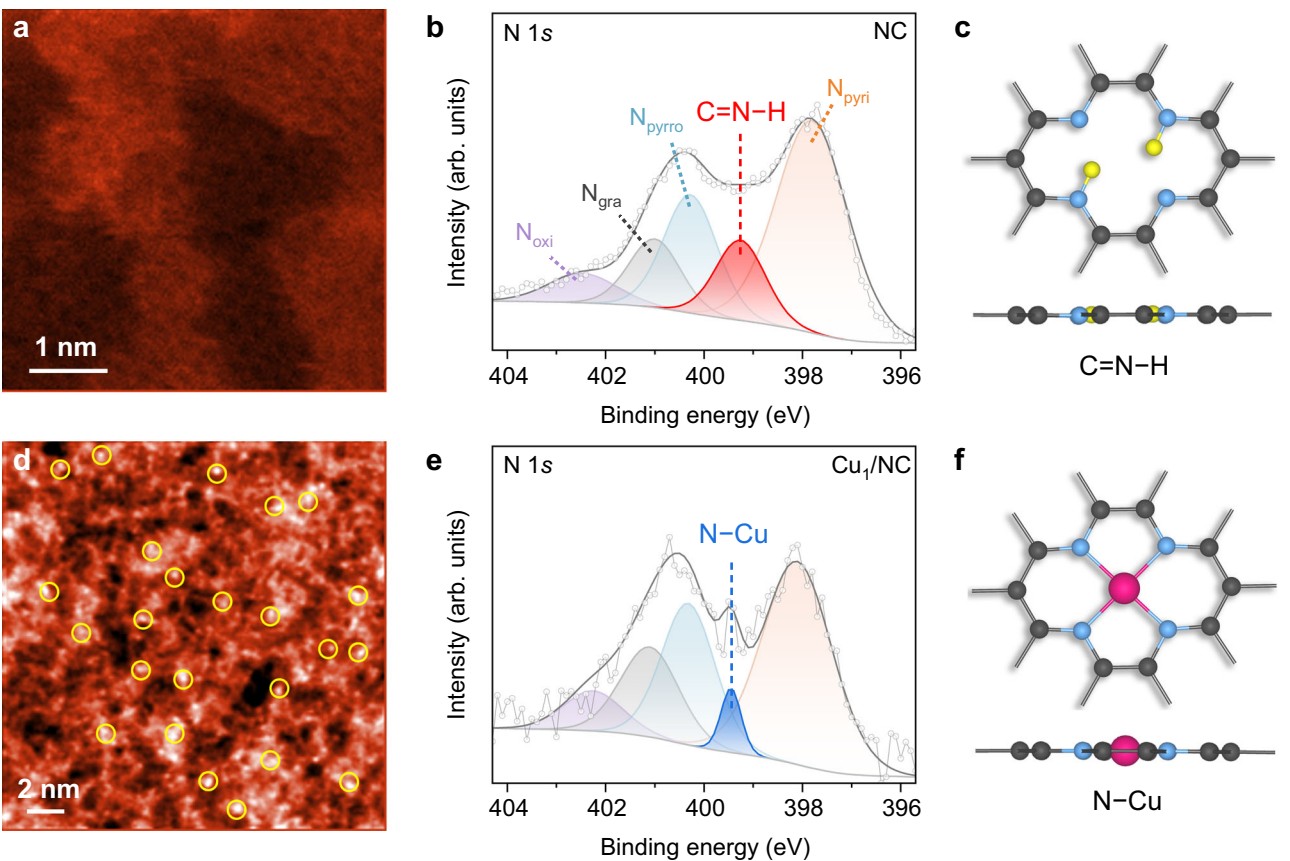

**Fig. 2 | Structural characterization of NC and Cu₁/NC. a**, **d** HAADF-STEM images of NC (**a**) and Cu₁/NC (**d**). **b**, **e** N 1*s* XPS spectra of NC (**b**) and Cu₁/NC (**e**). **c**, **f** Local structures of typical C=N–H species in NC (**c**) and N–Cu species in Cu₁/NC (**f**). Colour code: N, blue; C, black; H, yellow; Cu, pink.

Brunauer–Emmett–Teller (BET) plots, and a similar value holds for Cu₁/NC (Supplementary Figs. 12 and 13). We note that no obvious trace of $C_3N_4$ has been detected in all the FTIR and Raman spectra.

X-ray photoemission spectroscopy (XPS) was carried out to characterize the nitrogen and Cu species on NC and Cu₁/NC (Supplementary Figs. 14–16). The N 1*s* spectrum of NC (Fig. 2b) can be deconvoluted into components corresponding to pyridinic N ($N_{pyri}$, 397.9 eV), pyrrolic N ($N_{pyrro}$, 400.3 eV), graphitic N ($N_{gra}$, 401.0 eV), oxidized N ($N_{oxi}$, 402.5 eV) and C=N–H (399.3 eV)[47–50]. For the N 1*s* XPS spectrum of Cu₁/NC (Fig. 2e), a sharp peak emerges at 399.5 eV, which can be indexed to the N–Cu bonds[9,51,52], replacing the broad C=N–H peak. Accordingly, we may expect that the incorporation of Cu single atoms can suppress the N–H bonds at the pyridinic/pyrrolic N atoms in NC. These results coincide with the recent work[49] that demonstrated the dehydrogenation of nitrogen-doped carbon at the anchoring sites of transition-metal atoms during the synthesis of single-atom catalysts. We have performed density functional theory (DFT) calculations to assess the energetics of N–H bond formation on NC (Supplementary Fig. 17). A configuration of four neighbouring pyridinic N (denoted as N₄) in a graphene sheet, which is identical to the Cu-N₄ moiety[53] in Cu₁/NC (Fig. 2f) but without the presence of Cu, was taken as the model system for the study of C=N–H species. The binding of the first H atom to one of the N atoms is highly exothermic, with a free energy change ($\Delta E_{1H}$) of −0.72 eV at 0 V versus RHE. The second H preferentially adsorbs onto another N atom furthest from the former one, giving a free energy change ($\Delta E_{2H}$) of −0.62 eV. Since the successive binding of the next two H atoms to the remaining N atoms will consume substantial energy (1.16 and 1.23 eV, respectively), they are unlikely to be trapped at experimental conditions. Consequently, each pyridinic N₄ moiety will spontaneously capture up to two H atoms (Fig. 2c). Their reversible removal in

electrochemical reactions has constituted the premise of Faradaic pseudocapacitance for NC[36,37].

## Electrocatalytic performance for urea synthesis

The evaluation of electrocatalytic selectivity of NC and Cu₁/NC for urea synthesis was performed in an H-type cell (Supplementary Fig. 18) via the chrono-amperometry (CA) method. An electrolyte composed of 0.1 M KHCO₃ and 0.1 M KNO₃ was adopted, and high-purity CO₂ was continuously bubbled to maintain saturation during the electrolysis. The concentration of urea was measured by diacetyl monoxime method (Supplementary Fig. 19)[14]. Besides urea, a series of side products, including ammonia ($NH_3$), nitrite ($NO_2^-$), hydrazine ($N_2H_4$), carbon monoxide (CO) and hydrogen ($H_2$), were also identified through spectrophotometric and gas chromatographic analysis (Supplementary Figs. 20–22). The product yields at each potential were averaged over three independent measurements. As displayed in Fig. 3a and Supplementary Fig. 24, urea is the predominant product on NC catalyst from −0.3 to −0.5 V versus RHE. According to the molar yields of N-containing products, the N-selectivity was obtained, showing a relatively high proportion of $NO_3^-$ converted into $NO_2^-$. In spite of this, the FE of $NO_2^-$ product is rather limited as compared with that of urea, given the much smaller number of transferred electrons for conversion into $NO_2^-$ than urea. Moreover, the hydrogen evolution reaction (HER) is significantly suppressed on NC (Supplementary Fig. 25) as compared to those previously designed catalysts for urea electrosynthesis[10]. This feature represents one of the most critical attributes of the superior urea selectivity on NC.

In contrast, Cu₁/NC presents an FE of below 15% towards urea with excessive formation of side products (Fig. 3b), among which NtrRR products ($NO_2^-$ and $NH_3$) take the lead in the whole potential range investigated. Given the nearly identical synthesis procedure for both

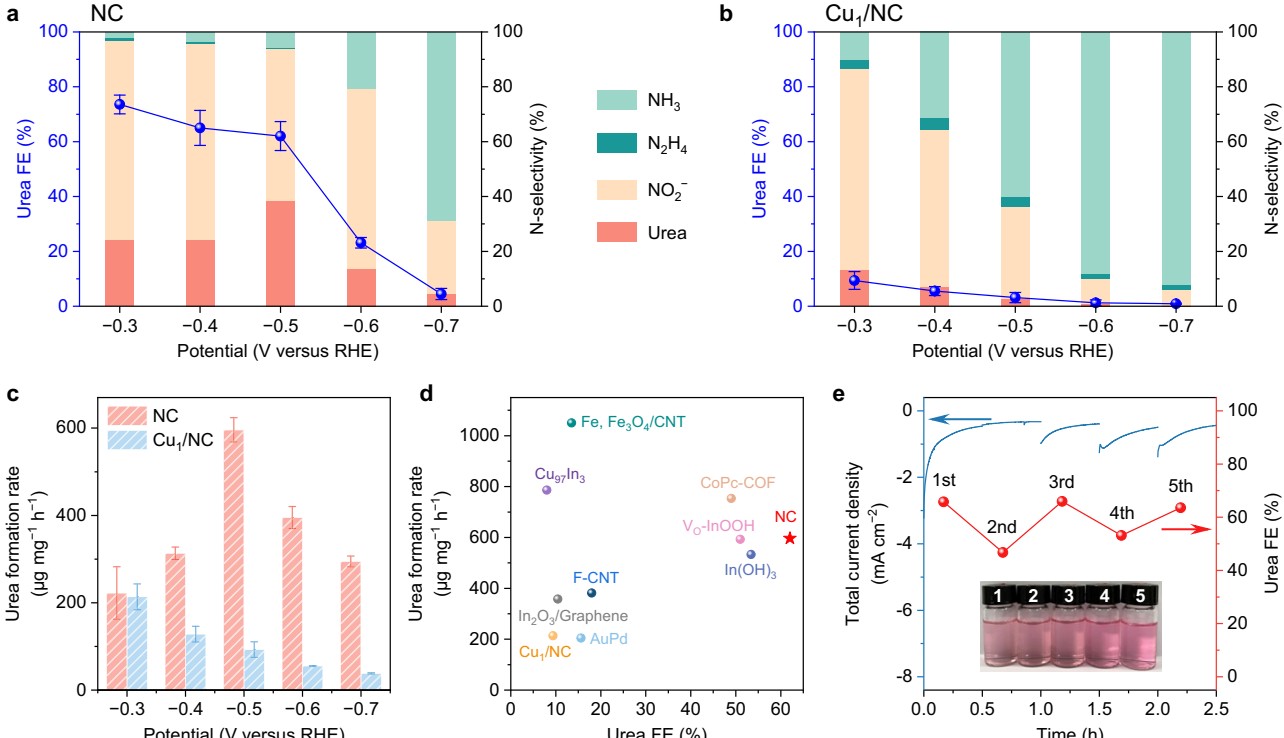

**Fig. 3 | Electrocatalytic performance of urea synthesis on NC and Cu₁/NC.**
**a**, **b** The FEs of urea and the N-selectivity of all N-containing products under different potentials on NC (**a**) and Cu₁/NC (**b**). **c** Urea yield rates at different potentials. **d** Comparison of urea formation rate between NC and other catalysts reported in the literatures, including Fe + Fe₃O₄ at carbon nanotubes (Fe, Fe₃O₄/CNT)[71], In(OH)₃[14], oxygen-deficient InOOH (V$_O$-InOOH)[72], F-doped carbon nanotubes (F-CNT)[73], Cu₉₇In₃[74], phthalocyanine-based covalent organic framework (CoPc-COF)[75], In₂O₃/Graphene[76], and AuPd[77]. Corresponding values are tabulated in Supplementary Table 1. **e** Stability test of NC at −0.5 V versus RHE for 5 cycles. The inset shows that the colour of the solution reacted with diacetyl monoxime is almost identical during cycling.

NC and Cu₁/NC, their major difference likely stems from the C=N−H and N−Cu species based on whether Cu ions were incorporated. This speculation was substantiated by the XPS results shown in Fig. 2b, e. Thus, we may postulate that the huge difference in the performance of urea synthesis should be attributed to the above two species, both of which could serve as the catalytic centres for the reduction of $NO_3^-$ and $CO_2$. Moreover, we have examined other single-atom catalysts, including In₁/NC and Fe₁/NC, to co-reduce $NO_3^-$ and $CO_2$ (Supplementary Fig. 26). Only negligible amounts of urea are formed for both catalysts due to the competitive NtrRR. The seesaw between $NO_3^-$ and $CO_2$ reduction is always tilted to the former, thus hindering C−N coupling, which appears to be a common situation in carbon-supported single-atom catalysts.

The effect of applied potentials on the yield rate of urea is displayed in Fig. 3c. As compared with Cu₁/NC, the NC catalyst produces considerably higher yields at −0.5 V versus RHE. Despite a larger total current density delivered by Cu₁/NC (Supplementary Fig. 27), it consumes most electrons to generate side products, resulting in remarkably poor selectivity to C−N coupling. C−N coupling is a potential-independent step, and when the bias potential is sufficiently negative, this reaction step would be kinetically less favourable than the excessive reduction of the intermediates into side products. This can rationalize the decreasing urea yields for NC at −0.5 to −0.7 V versus RHE. Notably, the NC catalyst enables urea electrosynthesis at a maximum yield rate of 596.1 μg mg⁻¹ h⁻¹ with a promising FE of 62% under −0.5 versus RHE, which is superior to most of the recently reported catalysts working at similar potentials (Fig. 3d). The durability test of NC showed almost no degradation in either activity or urea selectivity for 5 successive runs (Fig. 3e and Supplementary Fig. 28). TEM measurements were further performed after the test (Supplementary

Fig. 29), showing that the morphology of NC catalyst remains largely intact. The sustained electrocatalytic activity not only demonstrates the long-term stability of NC, but also indicates that the nitrogen and carbon sources originate from $NO_3^-$ and $CO_2$ rather than from the pyridinic/pyrrolic N and carbon atoms in the catalyst. In addition, we have synthesized another NC sample at an elevated pyrolysis temperature, which results in a greatly reduced amount of C=N−H species and exhibits a much lower urea FE in the co-reduction reaction (Supplementary Figs. 30 and 31). This implies that C=N−H plays a pivotal role for urea synthesis on NC.

## Control experiments for mechanistic rationalization

To better understand the electrocatalytic property of NC and Cu₁/NC, we performed control experiments to evaluate their activity for NtrRR and CO₂RR separately. In the electrochemical tests of individual NtrRR, argon gas instead of $CO_2$ was fed into the electrolyte. Over the NC catalyst, $NO_2^-$ was detected to be the major product at −0.3~−0.5 V versus RHE (Supplementary Fig. 34), while $NH_3$ formation predominated on Cu₁/NC with FE reaching nearly 100% at negative potentials exceeding −0.5 V versus RHE. We find that Cu₁/NC can deliver a much higher NtrRR current density (including the formation of $NO_2^-$, $N_2H_4$ and $NH_3$) than NC (Fig. 4a). Given that $NH_3$ is the most reduced product along the reaction path of NtrRR, the above results suggest that the replacement of C=N−H by N−Cu could promote the maximum reduction of $NO_3^-$. Intuitively, N−Cu may seem beneficial for urea electrosynthesis because the formation of urea (CO(NH₂)₂) also requires the maximum reduction of N in $NO_3^-$ to a valency of −3. However, as shown below, the urea selectivity would actually be adversely impacted by the facile NtrRR process when the CO₂RR process shows much inferior activity to NtrRR. Figure 4b and

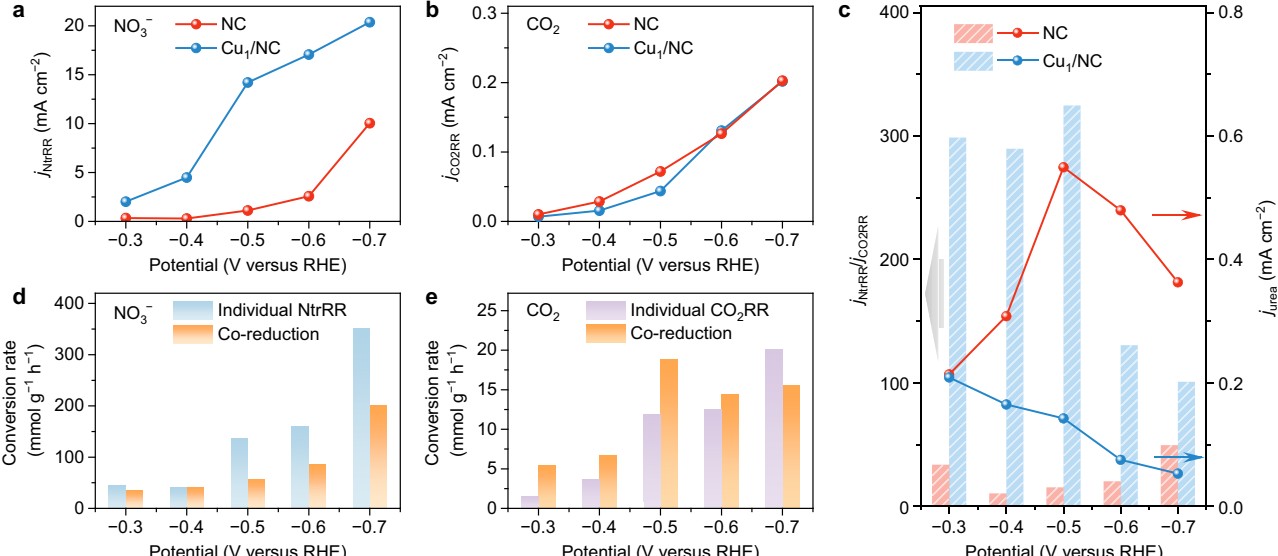

**Fig. 4 | Control experiments of individual NtrRR and individual CO₂RR on NC and Cu₁/NC. a** NtrRR current density ($j_{NtrRR}$, corresponding to the sum of partial current densities of $NO_2^-$, $N_2H_4$ and $NH_3$) during the individual reduction of $NO_3^-$. **b** CO₂RR current density ($j_{CO2RR}$, corresponding to the partial current density of CO) during the individual reduction of $CO_2$. **c** The ratio between $j_{NtrRR}$ and $j_{CO2RR}$, and the urea partial current density ($j_{urea}$) during the co-reduction of $NO_3^-$ and $CO_2$. **d** The conversion rate of $NO_3^-$ to N-containing products on NC in individual NtrRR as compared with co-reduction reaction. **e** The conversion rate of $CO_2$ to C-containing products on NC in individual CO₂RR as compared with co-reduction reaction.

Supplementary Fig. 35 present the results of individual CO₂ reduction experiments, in which $NO_3^-$ was removed from the electrolyte and the catalysts were run under a CO₂ gas flow. While the derived CO₂RR current densities are similar between NC and Cu₁/NC, they are one order of magnitude smaller than the NtrRR current densities in individual $NO_3^-$ reduction. DFT calculation results corroborate this finding, revealing that the first hydrogenation step of CO₂, i.e., *CO₂ → *COOH, is rate-determining and strongly endothermic (>1.3 eV) for both NC and Cu₁/NC (Supplementary Fig. 36). With the balance between NtrRR and CO₂RR struck during co-reduction of $NO_3^-$ and CO₂, there would be limited opportunity for urea formation at a single catalytic centre if the activity of this site remains unchanged in the reactions.

Cu₁/NC conforms to the above scenario, which is best seen when the potential becomes more negative. On Cu₁/NC, activity for both individual NtrRR and individual CO₂RR are obviously promoted from −0.3 to −0.7 V versus RHE (Fig. 4a, b), whereas the yield of urea shows a descending trend (Fig. 3c). A fiercer competition between both reactions at more negative potentials engages the co-reduction process into a more imbalanced state. The predominating NtrRR process, which takes more advantages of the bias potential than the potential-independent C−N coupling process, would thereby significantly inhibit urea formation. Unlike Cu₁/NC, the NC catalyst can steer the co-reduction of $NO_3^-$ and CO₂ along the urea formation path. We show in Fig. 4c that as compared to Cu₁/NC, the ratio between NtrRR and CO₂RR current densities in the control experiments is considerably smaller. Obviously, NtrRR on NC occupies a less predominating position than that on Cu₁/NC, which could offer more chance for C−N coupling during the co-reduction reaction. Another point worth mentioning is that on NC, the conversion rate of CO₂ as obtained from the yield rates of urea and CO (in mmol g⁻¹ h⁻¹) in co-reduction surpasses that in individual CO₂RR from −0.3 to −0.6 V versus RHE (Fig. 4e). This clearly suggests that the electrochemical conversion of CO₂ could be activated in the presence of $NO_3^-$. According to DFT calculations, we can rule out the possibility of C−N coupling between CO₂ and the main intermediates of NtrRR on NC (Supplementary Fig. 37). Hence, the hydrogenation of CO₂, especially the rate-determining *CO₂ → *COOH step in CO₂RR, is required for urea

formation. In this context, the promoted CO₂ conversion would imply that the NtrRR process plays a nontrivial role in reducing energy consumption at this elementary step. Such reduction can be interpreted as an NtrRR-induced alteration of the catalyst activity.

Notably, we find that on NC catalyst, the conversion rate of $NO_3^-$ is either minimally affected or considerably reduced upon the introduction of CO₂ as a reactant (Fig. 4d). It means that the NtrRR intermediates can participate in the urea formation process but seems not able to be activated from the CO₂RR intermediates. NtrRR is even hindered by the presence of CO₂ to some extent. Considering that the onset potential of NtrRR is more positive than that of CO₂RR (Supplementary Fig. 38), it can be safely inferred that NtrRR precedes CO₂RR during urea synthesis. In this context, $NO_3^-$ is first reduced at the catalytic centre and the reaction proceeds to $NO_2^-$ via two hydrogenation steps (valency of N changing from +5 to +3). Since the FE of $NO_2^-$ in individual NtrRR is nearly 100% at −0.3-−0.4 V versus RHE (Supplementary Fig. 34), only a negligible amount of $NO_2^-$ can be further reduced in this potential range, which can eliminate the possibility that the reduction products beyond $NO_2^-$ could participate in C−N coupling during co-reduction. This suggests that the switch of reaction path to CO₂RR is most likely accomplished in the period of forming $NO_2^-$. Subsequently, CO₂RR occurs in the vicinity of the previously formed NtrRR intermediates/products, and the CO₂RR intermediates can readily combine with these N-containing species to form urea. The above considerations have prompted us to propose that the co-reduction of $NO_3^-$ and CO₂ on NC adopts a sequential model via a consecutive switch between two reduction reactions.

## Operando tracking of the surface species

Operando attenuated total reflection surface-enhanced infrared absorption spectroscopy (ATR-SEIRAS) can provide critical details to better understand the $NO_3^-$ and CO₂ co-reduction mechanism on NC catalyst. Figure 5a shows the ATR-SEIRAS spectra with applied potentials varying from 0 to −0.7 V versus RHE, where several infrared bands are detected in the wavenumber ranges of 1000−2000 and 2800−3800 cm⁻¹. Two bands at around 1625 and 3370 cm⁻¹ could be

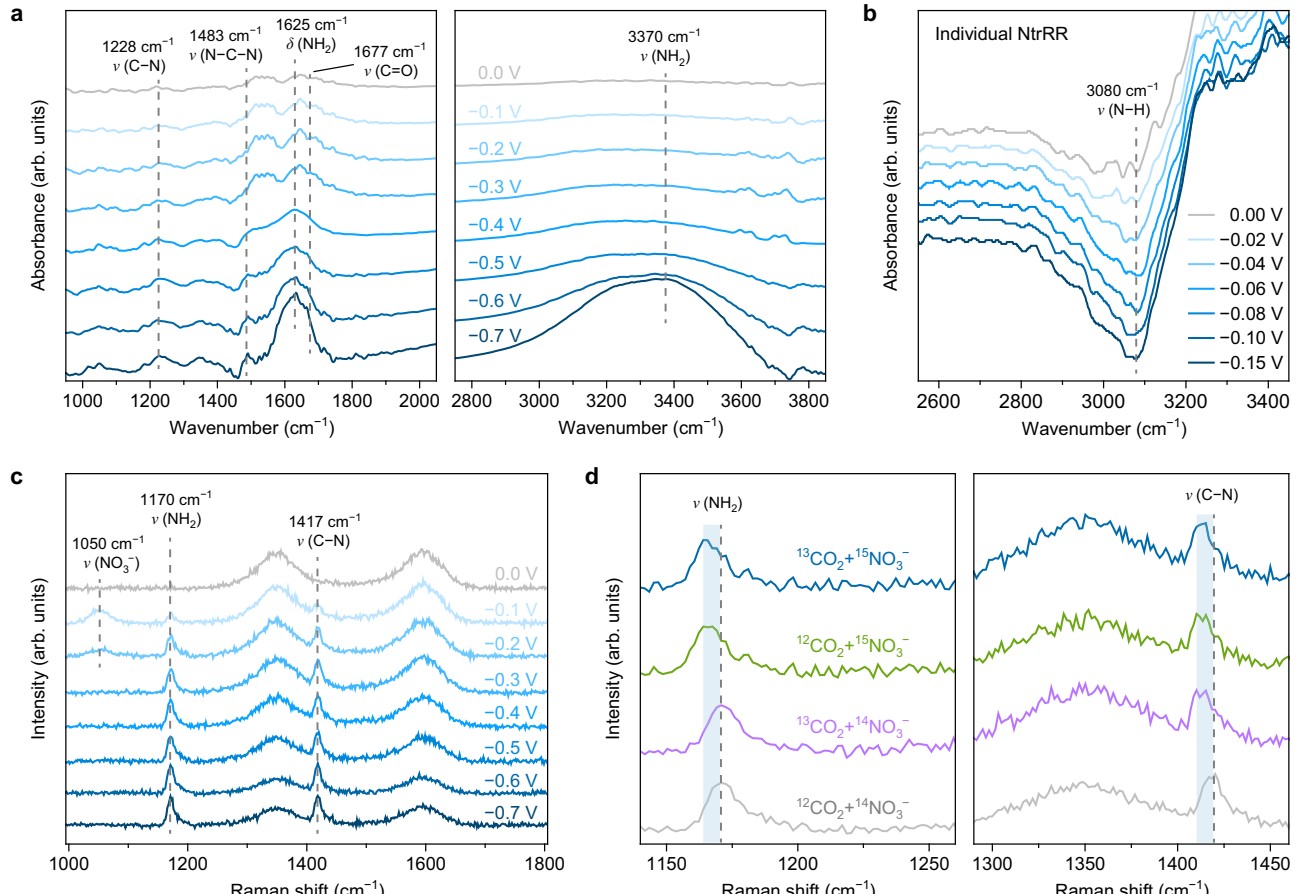

**Fig. 5 | Operando attenuated total reflection surface-enhanced infrared absorption spectroscopy (ATR-SEIRAS) and isotope-labelled in situ Raman characterization. a** ATR-SEIRAS spectra for NC under different applied potentials during co-reduction of $CO_2$ and $NO_3^-$. **b** ATR-SEIRAS spectra for NC in the range of relatively low applied potentials during individual NtrRR. **c** In situ Raman spectra for NC under different applied potentials during co-reduction of $CO_2$ and $NO_3^-$. **d** Comparison of the in situ Raman spectra under isotope-labelled $NO_3^-$ and $CO_2$ at −0.50 V versus RHE.

assigned to the bending and stretching (including symmetrical and antisymmetrical components) modes of −$NH_2$, respectively[14,54], both of which show a marked increase in intensity from −0.5 to −0.7 V. This is consistent with the surge in production of $NH_3$ at voltages approaching −0.7 V as revealed in electrochemical experiments. The bands centred at around 1483 and 1288 $cm^{-1}$ correspond to the antisymmetrical N−C−N stretching vibration of urea and the C−N stretching vibration of the reaction intermediates, respectively[55,56], while the stretching mode of C=O is also observed at ~1677 $cm^{-1}$ [57,58]. Their intensity steadily grows at voltages from 0 to −0.5 V and is modestly evolving afterwards, which reflects the successful C−N coupling and the production of urea at this voltage range during the co-reduction of $NO_3^-$ and $CO_2$.

To elucidate the changes in catalytic centre on NC during the reduction reactions, we investigate the evolution of N−H species in individual NtrRR by means of operando ATR-SEIRAS. To minimize the influence from $NH_3$/$N_2H_4$ products, the potential is scanned from 0 to −0.15 V; at more negative voltages the formation of $NH_3$/$N_2H_4$ will be initiated. We stress that while positive peaks in an ATR-SEIRAS spectrum represent an increase in concentration of the corresponding species at or near the surface, negative peaks can be ascribed to a decrease in concentration of the existing species. As shown in Fig. 5b, a negative peak appears at 3080 $cm^{-1}$ that is attributed to N−H stretching vibration[59-62], and its intensity is enhanced at more cathodic potentials. This suggests that the protons on existing C=N−H species at NC catalyst are diminishing at an initial stage of NtrRR, consistent with previous studies demonstrating the easy cleavage of N−H bonds on

NC[36,37]. Therefore, we can infer that the NtrRR process directly consumes the protons on C=N−H species. Such a change in the catalytic centre is expected to produce an alteration in activity, thus creating the opportunity to switch the reaction path to $CO_2$RR and enable the sequential mechanism for $NO_3^-$ and $CO_2$ co-reduction to urea. After the formation of urea, the high thermodynamic driving force as mentioned above ($\Delta E_{2H}$) could trigger rehydrogenation of NC, and the C=N−H species can serve as dynamically evolving active centres in the co-reduction reaction. We note that this N−H depletion feature is barely discernible on $Cu_1$/NC in operando ATR-SEIRAS (Supplementary Figs. 39 and 40).

In situ Raman spectroscopy was also conducted to monitor the surface species during the catalytic reaction. As shown in Fig. 5c, a peak at 1050 $cm^{-1}$ corresponding to the symmetrical stretching mode of nitrate ion[63,64] is observed. The peaks that appeared at 1170 and 1417 $cm^{-1}$ could be assigned to the stretching vibration of −$NH_2$ and C−N, respectively[65,66], indicating their formation in the co-reduction of $NO_3^-$ and $CO_2$. $^{15}N$-isotope and $^{13}C$-isotope-labelled experiments (Fig. 5d) were performed under an applied potential of −0.5 V versus RHE with $^{15}NO_3^-$ and $^{13}CO_2$ as the electrolyte and feeding gas. The positions of D and G bands of NC remain unchanged regardless of the isotope used. The Raman peak of −$NH_2$ band shows a red shift in $^{15}NO_3^-$ isotope substitution experiments, and the peaks of $^{12}C-^{15}N$, $^{13}C-^{14}N$, $^{13}C-^{15}N$ are evidently shifted to lower values as compared to $^{12}C-^{14}N$. These results verify that both nitrogen and carbon sources of urea originate from the employed feedstocks rather than from the functional groups on NC catalyst.

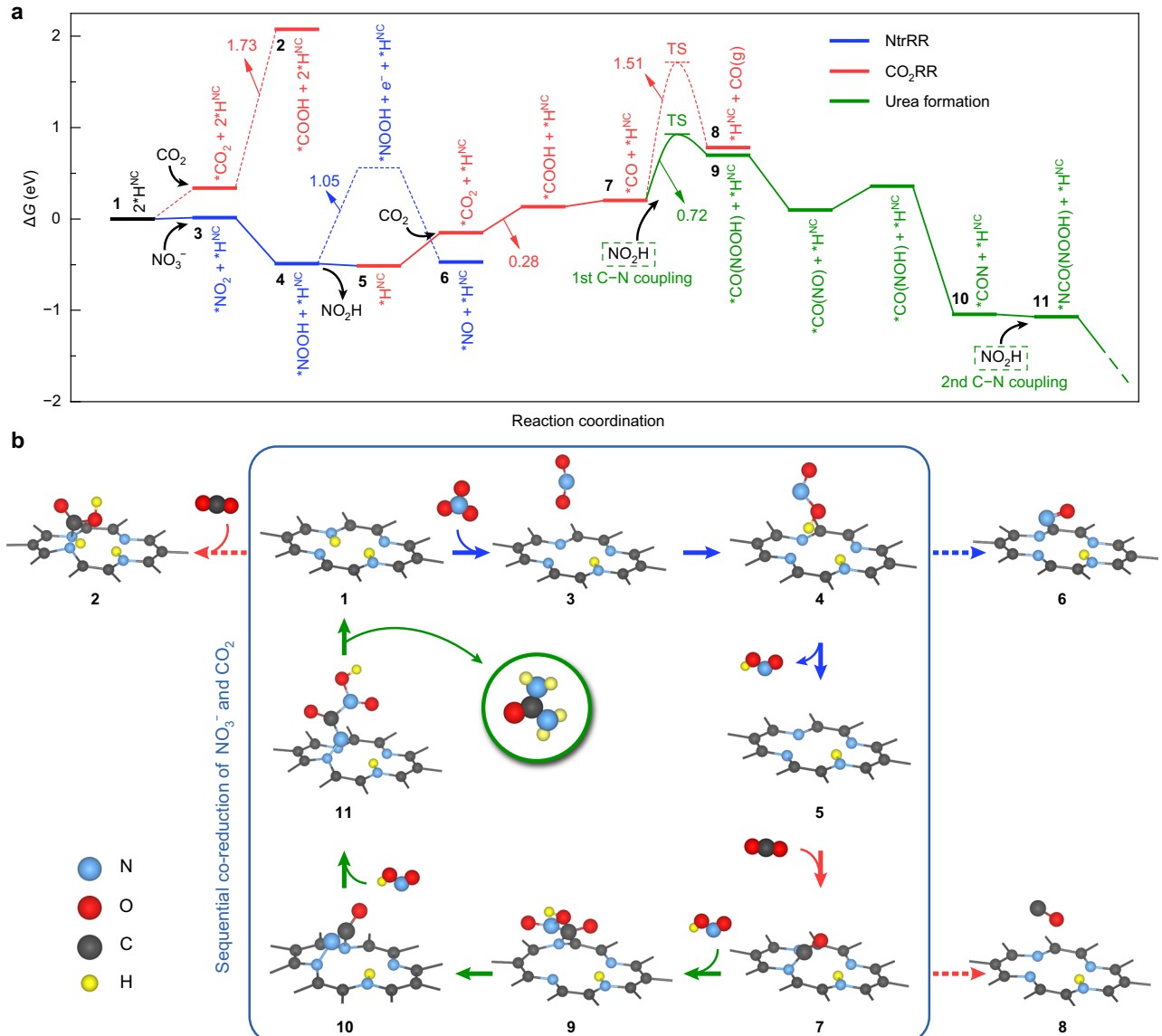

**Fig. 6 | Proposed reaction path of the sequential co-reduction reaction on NC.**
**a** The DFT-calculated Gibbs free energy profile of the co-reduction reaction of $NO_3^-$ and $CO_2$ at 0 V versus RHE. Elementary steps marked by dashed lines are less favourable than those marked by solid lines. The transition states (TS) of *CO

desorption (**7** → **8**) and C−N coupling (**7** → **9**) were obtained by CI-NEB calculations. **b** Structural configurations of the critical reaction intermediates. Intermediates in the frame are related to the sequential co-reduction mechanism leading to urea formation.

## DFT calculations of sequential reduction mechanism

Leveraging DFT Gibbs free-energy calculations, we can address the reaction pathway of urea electrosynthesis on the NC catalyst, which lends strong support to the sequential reduction scenario. Hydrogenated pyridinic $N_4$ site with two bonded H atoms (configuration **1** in Fig. 6, denoted as $2*H^{NC}$) was taken as the representative catalytic centre. At the beginning, NtrRR will readily take place while $CO_2RR$ is disfavoured because of the significant barrier to form *COOH (**2**). Although $NO_3^-$ can be reduced into *$NO_2$ in an electrochemical step ($NO_3^- + 2H^+ + e^- \rightarrow *NO_2 + H_2O$), this conversion can also occur via proton-coupled electron transfer from the catalyst ($NO_3^- + *H^{NC} \rightarrow *NO_2 + OH^-$). In other words, $NO_3^-$ derives a proton-electron pair from the dehydrogenation of the $N_4$ site, thus leaving one $H^{NC}$ on the $N_4$ site and forming a configuration of *$NO_2 + *H^{NC}$ (**3**), which is endothermic by only 0.08 eV. We find that the *$NO_2$ intermediate cannot chemically bind to the catalyst, but is in fact trapped by van der Waals forces and located at a distance of around 3 Å from the catalytic centre (Supplementary Fig. 41). Similarly, *NOOH (**4**) and *NO (**6**) are

physically rather than chemically adsorbed to the catalyst. The free-standing nature of these intermediates has led to a low overlap of electronic states between the adsorbed molecules and the catalyst. In particular, an energy gap of 1.05 eV exists between the lowest unoccupied molecular orbital (LUMO) of *NOOH and the Fermi level in configuration **4** (Supplementary Fig. 42). This result indicates that substantial energy is required to inject an electron from the catalyst to the adsorbed *NOOH molecule, thus imposing severe constraints on its electrochemical reduction. In comparison, the desorption process of *NOOH (**4**) is energetically favourable, exothermic by −0.05 eV, which offers ample opportunity for exposing the $N_4$ site with one $H^{NC}$ atom remaining (**5**). The easy desorption and migration of $NO_2H$ found by DFT calculations are in line with the experimental observation that NC catalyst is selective towards $NO_2^-$ formation during individual reduction of $NO_3^-$.

Subsequently, from configuration **5**, the reduced steric effect at the catalytic centre due to a missing $H^{NC}$ is at play to guarantee facile $CO_2RR$ and C−N coupling processes. The steric effect gives rise to a

strong interaction between *COOH and NC catalyst (Supplementary Fig. 43), which is the reason for the remarkable reduction in energy consumption at the $*CO_2 \rightarrow *COOH$ step with one $H^{NC}$ (0.28 eV) as compared to that with two $H^{NC}$ atoms (1.73 eV). The high propensity for $CO_2RR$ at this stage supports our speculation that $CO_2$ conversion can be promoted after NtrRR. The next bifurcation of the reaction pathway occurs at *CO, which will undergo either desorption or C−N coupling. Since both steps are kinetics-dominated processes, we here employed climbing-image nudged elastic band (CI-NEB)[67] calculations (Supplementary Fig. 44) to determine their activation barriers. The results show that the barrier for the formation of CO(g) (**8**) is substantially higher than that of *CO(NOOH) (**9**), well matching with the inferior selectivity to CO product in the experiments. Then, the *CO(NOOH) intermediate undergoes three sequential electrochemical reduction steps and releases two water molecules to form *CON (**10**), at which the second C−N coupling takes place utilizing another $NO_2H$ nearby. The resultant intermediate *NCO(NOOH) (**11**) will undergo reduction into urea, and all the elementary steps in between are thermodynamically spontaneous (Supplementary Fig. 45) except for the $*NCO(NO) \rightarrow *NCO(NOH)$ step (0.34 eV). After the urea molecule leaves the catalyst surface, the $N_4$ site will be hydrogenated immediately, leading to the regeneration of the $2*H^{NC}$ (**1**) configuration (Supplementary Fig. 17).

Collectively, the DFT results in Fig. 6 provide us with a comprehensive picture of the sequential reduction mechanism on NC. At the core of this mechanism lies the capability of structural alteration of the catalytic centre (between **1** and **5**), which is dynamic and reversible in the urea formation process. For comparison, DFT calculations were also performed for $Cu_1/NC$. The $Cu$-$N_4$ moiety corresponds to the catalytic centre, which may not undergo hydrogenation due to the saturation of all N atoms. The NtrRR process is initiated by the electrochemical reduction of $NO_3^-$, followed by further reduction into *NO with a flattened energy landscape (Supplementary Fig. 48). We note that unlike on NC catalyst, all the intermediates on $Cu_1/NC$ are chemically bonded to the catalytic centre (Supplementary Fig. 49), thus avoiding the exposure of the catalytic centre to other reactants. NtrRR continues to proceed until the formation of $NH_3$ is completed, with a limiting potential of 0.31 V. On the other hand, the $CO_2RR$ process is sluggish due to the highly endothermic $*CO_2 \rightarrow *COOH$ step (1.59 eV), similar to the case of $N_4$ site with two $H^{NC}$ atoms. As the $Cu_1/NC$ catalyst remains intact throughout the reactions, an equally huge amount of free energy input would be required to activate this step and afford the *CO intermediate, which is indispensable for urea formation. This suggests that $NH_3$ is the most accessible product in the co-reduction of $NO_3^-$ and $CO_2$ on $Cu_1/NC$, showing good agreement with the experimental results. The proposed rationalization may also provide an explanation for the change in product selectivity on $Cu_1/NC$: the overall >90% $NH_3$ selectivity observed experimentally at −0.7 V is correlated with the competition for catalytic centres between NtrRR and $CO_2RR$, while less negative potentials would lead to less fierce competition, and therefore more ample opportunities for concomitant reduction of both reactants.

## Discussion

In this work, selective urea electrosynthesis via a sequential reduction process utilizing $NO_3^-$ and $CO_2$ as reactants is proposed and demonstrated on an N-doped carbon catalyst. This NC catalyst delivers a urea yield rate of 596.1 $\mu$g $mg^{-1}$ $h^{-1}$ with a high FE of 62% at −0.5 V versus RHE, outperforming most of the previously reported catalysts and showing great potential in large-scale application. In sharp contrast with NC, the $Cu_1/NC$ single-atom catalyst displays a much lower urea selectivity, which originates from the fierce and constant competition between the reduction of $NO_3^-$ and $CO_2$ at a single catalytic centre. Based on rationally designed experiments, operando measurements and DFT calculations, we reveal the essential role of the C=N−H species

on NC that could serve as dynamic active sites. The reaction preference is inherently controlled by the number of N−H bonds and is switchable between favouring NtrRR process or $CO_2RR$ process, whereby the reduction reactions are sequential and steered towards urea formation. The fundamental understanding of the sequential reaction model can form the basis for furthering the development of selective electrocatalysts to permit facile C−N coupling and efficient synthesis of urea.

## Methods

### Synthesis of NC and $Cu_1/NC$

NC was prepared via a two-step pyrolysis method, using glucose (GC) and dicyandiamide (DCDA) as carbon source and nitrogen source, respectively. GC and DCDA were purchased from Sigma-Aldrich, and were directly used without further treatment. GC was mixed with DCDA at a mass ratio of 1: 40, and the mixture was sintered under flowing Ar at 550 °C for 6 h to form $C_3N_4$. Subsequently, the resultant product was carbonized at 900 °C for 3 h in an Ar atmosphere. The process of preparing $Cu_1/NC$ and other single-atom catalysts was the same as that of NC, except for adding the metal precursors such as $Cu(NO_3)_2$. Finally, aggregates of metal particles were removed by washing the catalysts thoroughly in 4.0 M $H_2SO_4$.

### Ex situ and in situ characterizations

The XRD patterns were collected on a Bruker D8 Advance diffractometer equipped with Cu Kα radiation, scanning from 10° to 80° with a scan rate of 2° $min^{-1}$. The structure and morphology of the samples were investigated on a field emission scanning electron microscope (ZEISS SUPRA®55, Carl Zeiss) and a transmission electron microscope (JEM−3200FS, JEOL) equipped with an EDS detector. XPS experiments were conducted on an ESCALAB 250X instrument (Thermo Fisher).

The electrochemical operando ATR-SEIRAS was measured by INVENIO R FTIR spectrometer (Bruker) equipped with a mercury-cadmium-telluride (MCT) detector. An Ag/AgCl electrode and a Pt foil were employed as the reference electrode and the counter electrode, respectively. During spectrum collection, the optical path was continuously purged with nitrogen gas to minimize the disturbance caused by water and $CO_2$ in the air. A $CO_2$-saturated electrolyte containing 0.1 M $KHCO_3$ and 0.1 M $KNO_3$ was employed in the co-reduction reaction process of $NO_3^-$ and $CO_2$. The background spectrum of the catalyst electrode was obtained at an open-circuit potential before each measurement. Then, the absorbance spectra ($-\log(R/R_0)$) at different potentials were collected at a spectral resolution of 4 $cm^{-1}$. In the ATR-SEIRAS spectra, a negative peak indicates that a certain substance or functional group is consumed, while a positive peak indicates that a certain substance or functional group is produced or has increased.

The in situ Raman characterization was performed in a three-electrode cell equipped with a confocal microscope Raman system (Renishaw inVia). The wavelength was 532 nm with a 50× microscope objective. During the testing process, the distance between the sapphire window and the electrode was less than 0.1 nm, which ensures that the attenuation effect of the solution layer on Raman signal can be as small as possible.

### Electrochemical measurement

The electrochemical measurement was conducted on a three-electrode configuration using a CHI 660E electrochemical station in the H-type cell. Nafion 115 (Dupont) was pretreated and used to assemble the H-type cell. The Nafion membrane was first heated in 5% $H_2O_2$ at 80 °C for 1 h and rinsed by deionized water. Then, it was further heated in 5% $H_2SO_4$ at 80 °C for 1 h and rinsed by deionized water again. An electrolyte composed of 0.1 M $KHCO_3$ + 0.1 M $KNO_3$ and saturated with $CO_2$ (pH = 6.8) was employed, with an Ag/AgCl

electrode (in saturated KCl) and a platinum foil (1 cm × 1 cm) used as the reference electrode and the counter electrode, respectively. 1 mg of catalyst material was dispersed in 900 μL deionized water, 50 μL isopropanol and 50 μL Nafion (5 wt% aqueous solution), and then sonicated for 3 h under ice-water bath. Afterwards, 150 μL of the catalyst ink was loaded onto a carbon paper (Toray, TGP-060) and dried in the ambient environment to form the working electrode (geometric area: 1 × 1 cm²; mass loading: 0.15 mg cm⁻²). Before electrochemical tests, $CO_2$ was purged into the electrolyte for 30 min with a flow rate of 100 sccm to remove the residual air. Potentiostatic electrolysis for 30 min with a $CO_2$ flow rate of 20 sccm was done at each potential, and the gas products were analysed with a 10-min interval by gas chromatography (SHIMADZU, GC-2014C) equipped with a flame ionization detector (FID) and a thermal conductivity detector (TCD). The gas chromatography was calibrated using standard samples under standard conditions (1 atm, 298 K). The linear sweep voltammetry test was performed at a scan rate of 10 mV s⁻¹. The N-selectivity of reaction products was calculated as follows:

$$S_{N-selectivity} = \frac{M(N)}{M_{total}(N)} * 100\% \tag{1}$$

where $M(N)$ and $M_{total}(N)$ are the moles of nitrogen for a specific N-containing product and for all products in NtrRR, respectively.

### Computational details

All the theoretical calculations were performed via spin-polarized DFT using Vienna ab initio simulation pack (VASP 5.3.5) with projector augmented wave method[68]. The Perdew−Burke−Ernzerhof (PBE) exchange-correlation functional was employed along with a plane-wave cutoff energy of 450 eV. The third-generation (D3) semi-empirical van der Waals corrections proposed by Grimme[69] were adopted in structural optimization to deal with the dispersion interactions. Supercells consisting of 5 × 5 unit cells were constructed for the carbon layers of both NC and Cu₁/NC catalysts, with a vacuum space of >20 Å to separate the slabs in the $z$ direction. The Brillouin zone was sampled by Γ-centred Monkhorst–Pack $k$-point mesh of 2 × 2 × 1 for free energy calculations. To take into account the contribution of hydrogen bonds, we introduced four water molecules in our simulations for the adsorbate configurations. Each image in the climbing-image nudged elastic band (CI-NEB) calculations was relaxed until a force tolerance of 0.02 eV/Å was reached.

Computational hydrogen electrode (CHE) model[70] was employed for Gibbs free energy calculation in every elementary step. In this scheme, the free energy of H⁺/e⁻ pair is equivalent to the chemical potential of gaseous $H_2$ at standard conditions (pH = 0, $P$ = 1 bar, $T$ = 298 K). The free energy change $\Delta G$ of each elementary step can be calculated by the following equation:

$$\Delta G = \Delta E + \Delta E_{ZPE} - T\Delta S + eU + k_B T \ln(10) \times pH \tag{2}$$

where $E$ is the total energy directly calculated by DFT; $E_{ZPE}$ is the zero-point energy, which was obtained from the calculated vibrational frequencies of the adsorbates using normal mode analysis; $S$ is the entropy and $T$ was set to room temperature (298 K); $U$ is the potential measured against the standard hydrogen electrode; $k_B$ is Boltzmann constant. The entropy for each reaction intermediate was calculated by the following equation:

$$S = \sum_i \frac{\frac{h\nu_i}{T}}{e^{\frac{h\nu_i}{k_B T}} - 1} - k_B \sum_i \ln\left(1 - e^{\frac{-h\nu_i}{k_B T}}\right) \tag{3}$$

where $h$ and $\nu_i$ are Planck constant and the vibrational frequencies of the adsorbates, respectively. In this work, the free energy profiles were calculated at 0 V versus RHE and pH = 6.8.

## Data availability

The data generated or analysed during this study are included in this published article and its supplementary information files, and are available from the corresponding authors on reasonable request. Source Data are provided with this paper.

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

## Acknowledgements
The authors acknowledge financial support from the Guangdong Basic and Applied Basic Research Foundation (2020A1515110843 and 2023A1515011391), the National Natural Science Foundation of China (22109003, 22308322 and 52373223), the Natural Science Foundation of Shenzhen (JCYJ20190813110605381), R&D Project of State Grid Corporation of China (No. 5108-202218280A-2-439-XG), Sichuan Science and Technology Program (2023NSFSC0434) and the Major Science and Technology Infrastructure Project of Material Genome Big-science Facilities Platform supported by Municipal Development and Reform Commission of Shenzhen.

## Author contributions
Yang Li and S.L. conceived the idea and designed the experiments. Yang Li, H.L. and Q.X. synthesized all the materials and conducted the electrochemical tests. M.H. carried out the operando measurements. S.Z. conducted the density functional theory calculations. H.L., Haocong Yi, Haibin Yang, Z.M., Q.Z., Z.W.Y. and W.L. supported the characterizations and analysis. S.L. supervised the research. Yang Li, M.H. and S.L. wrote the manuscript with advice from Yuan Lin, S.X.D. and F.P. All authors participated in the discussion and agreed with the conclusions of the study.

## Competing interests
The authors declare no competing interests.
