## [Peer Review File · Nature Communications]

Sequential co-reduction of nitrate and carbon dioxide enables selective urea electrosynthesisREVIEWER COMMENTS

Reviewer #1 (Remarks to the Author):

This manuscript from Yang Li et al. reported a nitrogen-doped carbon catalyst that affords highly selective conversion to urea by sequentially reducing NO_3^- and CO_2 at a dynamic catalytic center. To alleviate the competition between concurrent NO_3^- and CO_2 reduction, this sequential mechanism avoids the predominance of nitrate reduction reaction along the whole reaction path. However, there are still some issues that require further clarification, and the manuscript cannot be accepted at the present stage. I present questions and comments below.

- 1) The PXRD patterns of NC and Cu1/NC in the range from 5 to 80° should be provided, because the diffraction peak intensity of the (100) lattice plane is influenced by the in-plane long-range order structure of C_3N_4 .
- 2) In Fig. S3, defect-rich C_3N_4 exposes N atoms of triazine units. Please testify that the N atoms on triazine units convert to pyridine N after calcination at 900 °C.
- 3) The vibration peaks in Fig. S9 are very inconspicuous. The IR spectrum of NC needs to be retested.
- 4) It is confusing how $^*\text{CO}(\text{NOOH})$ converts to $^*\text{CON}$ in Fig. 6.
- 5) As the authors pointed out in the electrocatalytic performance for urea synthesis, “Due to the much smaller number of required electrons to convert NO_3^- to NO_2^- as compared to urea formation, the FE of NO_2^- product is still limited.” Wouldn't the NO_2^- product be easier to produce with less number of electrons required for conversion? Why is it restricted?
- 6) As the authors indicated, “The sustained electrocatalytic activity not only demonstrates the long-term stability of NC, but also indicates that the nitrogen source and the carbon source originate from NO_3^- and CO_2 rather than from the pyridinic/pyrrolic N and carbon atoms in the catalyst.” Probing the nitrogen source and the carbon source with isotopic labeling or other experiments is more credible.
- 7) The peaks of Cu^+ and Cu^0 are similar and difficult to distinguish by Cu 2p XPS spectrum. Please provide Cu LMM spectra to further verify the presence of Cu^+ .

Reviewer #2 (Remarks to the Author):

Reviewer Comment to

Manuscript Number: NCOMMS-23-06455-T

Title: Sequential co-reduction of nitrate and carbon dioxide enables selective urea electrosynthesis

Summary:

Authors investigated a nitrogen-doped carbon catalyst which has been employed to sequentially reduce NO_3^- and CO_2 with a Faradaic efficiency of 62% and urea yield rate of 596.1 $\mu\text{g mg}^{-1} \text{h}^{-1}$. Further authors proposed that reversible hydrogenation on the nitrogen functional groups that mitigates the overwhelming reduction of reactants, minimizing the formation of side product and highlighted the sequential reduction of nitrate and CO_2 . This is an interesting “concept”. However, there are some critical questions or suggestions for this manuscript from my aspect, as follows.

Major comments:

(1) In the study, from figure S12 and S13 in supporting information, the surface area on NC is much larger than Cu-NC (close to 5 times higher on NC than Cu-NC), while the urea yield rate on NC is less than three times of that on Cu-NC. This indicates that the intrinsic catalytic activity on Cu-NC for urea production is still higher than the NC catalysts even though the Cu-NC has poor selectivity.

(2) In page 10, is it solid to claim: "Given that NC and Cu1/NC were synthesized using the same method with only difference in whether Cu ions were incorporated, both catalysts may share nearly identical functional groups except for the C=N-H and N-Cu species." by the Figure 2(b) and (e)? In Fig. 2(e) original gray line in the XPS indicates its structure is more complicated. Is that possible proves this by removing the Cu species from the Cu-NC catalysts and then conduct the co-reduction? Besides, during operating conditions (at applied potential at -0.3 to -0.7 V) is it possible that the Cu will leach out and form C=N-H bond?

(3) "The reaction preference is inherently controlled by the number of N-H bonds and is switchable between favouring NtrRR process or CO2RR process" is doubtful. The preference of Nitrate reduction over CO2 reduction during Co-reduction of CO2 and Nitrate is intrinsic due to a higher reduction potential of Nitrate where it happens at even positive potential while the CO formation from CO2 requires a more negative potential. The mismatch between these two reduction potentials leads to this low FE of urea. For NC catalyst, it has a higher overpotential to reduce the Nitrate to NH3, this might bring the reduction potential difference between CO2 and NO3- smaller, leaving the possibility of C-N coupling instead of all NO3- becomes NH3 which seems happening on Cu-N-C. This can be seen in Fig. 4(a) and (b). It seems NC is an extremely poor nitrate reduction catalyst where nitrate reduction starts at potential < -0.4 V and at this potential CO forms from CO2. All this makes "proton-involved dynamic catalyst evolution" questionable. And further the results on Fig. 5 are not sufficient to prove this "proton-involved dynamic catalyst evolution" for higher urea selectivity.

(4) Why some catalysts with high urea production yield in Table S1 are not included in Fig.3 (d)?

(5) In stability test, is "5 successive runs" sufficient? Have you run longer cycles? Comparing Fig.3(b) and Fig. S27, the XPS before and after stability test indeed has some difference. The scattering circle markers are so different, how confident we are can still claim "showing that the structure, morphology and composition of NC catalyst remain largely intact."?

(6) Do the authors do the similar analysis (Fig. 5(b)) on Cu-NC? Do authors think that is that also possible that the formation of N-H bond under reduction conditions on Cu-NC?

Minor comments

(1) In the computational details, "The Gibbs free energy in every elementary step was calculated by the computational hydrogen electrode (CHE) model." Please add appropriate citation for this model.

(2) Sentence in page 5: "..... which is much superior to the previously reported catalysts." Seems only faradaic efficiency is better while yield rate still can be comparable to some reported literatures (Table S1) So, authors need rewrite it a bit.

Response to Reviewer #1

Reviewer 1: This manuscript from Yang Li et al. reported a nitrogen-doped carbon catalyst that affords highly selective conversion to urea by sequentially reducing NO_3^- and CO_2 at a dynamic catalytic center. To alleviate the competition between concurrent NO_3^- and CO_2 reduction, this sequential mechanism avoids the predominance of nitrate reduction reaction along the whole reaction path. However, there are still some issues that require further clarification, and the manuscript cannot be accepted at the present stage. I present questions and comments below.

Response:

We sincerely appreciate the efforts made by the reviewer in carefully reviewing our manuscript and providing valuable feedback. In response to the reviewer's concerns regarding the scientific mechanism and certain conclusions, we have addressed these issues by incorporating additional experiments and further discussions in the revised manuscript. We have thoroughly considered the reviewer's comments, and we believe that the revisions have significantly improved the quality of our research. Please find below our point-by-point response, and we hope that the reviewer will find our revisions satisfactory.

Comment 1. The PXRD patterns of NC and Cu_1/NC in the range from 5° to 80° should be provided, because the diffraction peak intensity of the (100) lattice plane is influenced by the in-plane long-range order structure of C_3N_4 .

Response:

We would like to thank the reviewer for this insightful comment. It is worth noting that C_3N_4 will undergo complete thermolysis at around 750°C , as reported by previous studies (Li et al. *Angew. Chem. Int. Ed.*, **2012**, 51, 9689-9692; Yu et al. *Adv. Mater.*, **2016**, 28, 5080-5086). In the present work, the temperature will reach 900°C for the synthesis of both NC and Cu_1/NC , and therefore we believe that only negligible amount of C_3N_4 can remain after being heated at this temperature for 3 h.

To confirm this hypothesis, we have conducted the XRD measurements in the range from 5° to 80° . In the case of typical C_3N_4 , a peak is observed at 13° , which corresponds to the in-plane ordering of tri-s-triazine units. However, in both NC and Cu_1/NC samples in the present work, we observed the absence of any peak associated with C_3N_4 (**Fig. R1a**), indicating the complete conversion of C_3N_4 . To facilitate a direct comparison, we have included the XRD patterns of previously reported C_3N_4 samples (Li et al. *Angew. Chem. Int. Ed.*, **2012**, 51, 9689-9692; Du et al. *Nat. Commun.*, **2023**, 14, 2278) in **Fig. R1b** and **R1c**.

We have updated **Supplementary Fig. 4** as **Fig. R1a**, and added the following statement in the revised manuscript (Page 6):

No peak can be assigned to the in-plane long-range order structure of C_3N_4 for both samples.

Figure R1. (a) XRD patterns of NC and Cu₁/NC in the present work. (b-c) XRD patterns of C₃N₄, as extracted from previous reports (Li et al. *Angew. Chem. Int. Ed.*, **2012**, 51, 9689-9692; Du et al. *Nat. Commun.*, **2023**, 14, 2278). [Li, X.-H., Kurasch, S., Kaiser, U. and Antonietti, M. (2012), Synthesis of Monolayer-Patched Graphene from Glucose. *Angew. Chem. Int. Ed.*, 51: 9689-9692. <https://doi.org/10.1002/anie.201203207>. Copyright © 2012 WILEY-VCH Verlag GmbH & Co. KGaA, Weinheim] [Du, L., Gao, B., Xu, S. et al. Strong ferromagnetism of g-C₃N₄ achieved by atomic manipulation. *Nat Commun* 14, 2278 (2023). <https://doi.org/10.1038/s41467-023-38012-8>.]

Comment 2. In Fig. S3, defect-rich C₃N₄ exposes N atoms of triazine units. Please testify that the N atoms on triazine units convert to pyridine N after calcination at 900 °C.

Response:

We would like to thank the reviewer for this suggestion. As mentioned in our last response, previous studies have revealed that C₃N₄ will convert to N-doped carbon materials at 900 °C. In this work, we have used one-pot two-step pyrolysis method to prepare NC and Cu₁/NC. The reason of sintering at 550 °C to form C₃N₄ is that heating the reactants directly to higher temperatures will lead to considerable sublimation. Forming C₃N₄ at the first step will help derive a larger amount of products (i.e., NC and Cu₁/NC) than one-step reaction.

According to the reviewer's comment, we have performed more characterizations to testify that nearly all N atoms on the triazine units have undergone conversion. Notably, results of XRD, FTIR, Raman and XPS measurements can firmly support this conclusion:

- **XRD patterns:** No peak emerges at 13° corresponding to the in-plane ordering of tri-s-triazine units, as displayed in **Fig. R1**. Similar results have been reported in the literature (Zhang et al. *Angew. Chem. Int. Ed.*, **2010**, 49, 441-444; Li et al. *Angew. Chem. Int. Ed.*, **2012**, 51, 9689-9692; Yu et al. *Adv. Mater.*, **2016**, 28, 5080-5086).
- **FTIR spectra:** No band is detected at 813 cm⁻¹ (**Fig. R2a**), which corresponds to the vibration of the triazine units in C₃N₄. To allow direct comparison, here we have extracted a typical FTIR spectrum of C₃N₄ (**Fig. R2b**) from the previous study (Zhao et al. *ACS Appl. Nano Mater.*, **2016**, 8, 21555-21562).

- **Raman spectra:** No band is detected at 978 cm^{-1} (**Fig. R3a**), which corresponds to the breathing mode of the triazine units in C_3N_4 . In **Fig. R3b**, we show a typical Raman spectrum of C_3N_4 as extracted from the previous study (Liu et al. *J. Chem. Eng.*, **2021**, 425, 130615).
- **XPS spectra:** The C $1s$ spectrum of the NC sample can be deconvoluted into components of C=C, C-N and C-O (**Fig. R4a**), while for C_3N_4 , a predominant peak emerges for N=C-N (**Fig. R4b**, extracted from Du et al. *Nat. Commun.*, 2023, 14, 2278). The large difference between both XPS spectra indicate a negligible amount of C_3N_4 in the NC sample.

Figure R2. (a) FTIR spectra of NC and Cu_1/NC . (b) FTIR spectrum of C_3N_4 , as extracted from a previous report (Zhao et al. *ACS Appl. Nano Mater.*, **2016**, 8, 21555-21562). [Reprinted (Adapted) with permission from Yafei Zhao, Ruoyan Wei, Xin Feng, Lining Sun, Panpan Liu, Yongxiang Su, and Liyi Shi, *ACS Appl. Mater. Interfaces* 2016, 8, 33, 21555–21562, DOI: 10.1021/acsami.6b06254. Copyright © 2016 American Chemical Society.]

Figure R3. (a) Raman spectrum of NC. (b) Raman spectrum of C_3N_4 , as extracted from a previous report (Liu et al. *J. Chem. Eng.*, **2021**, 425, 130615). [Reprint from Wei Liu, Chenjie Song, Mingpu Kou, Yongye Wang, Yu Deng, Toshihiro Shimada, Liqun Ye, Fabrication of ultra-thin g- C_3N_4 nanoplates for efficient visible-light photocatalytic H_2O_2 production via two-electron oxygen reduction, *Chemical Engineering Journal*, Volume 425, 2021, 130615, ISSN 1385-8947, <https://doi.org/10.1016/j.cej.2021.130615>. Copyright (2021), with permission from Elsevier.]

Figure R4. (a) C 1s XPS spectrum of NC. (b) C 1s XPS spectrum of C₃N₄, as extracted from a previous report (Du et al. *Nat. Commun.*, 2023, 14, 2278). [Du, L., Gao, B., Xu, S. et al. Strong ferromagnetism of g-C₃N₄ achieved by atomic manipulation. *Nat Commun* 14, 2278 (2023). [https://doi.org/10.1038/s41467-023-38012-8.](https://doi.org/10.1038/s41467-023-38012-8)]

Figure R2a, R3a and R4a are all provided in the revised Supplementary Information. The original Supplementary Fig. 9 has now been replaced by Fig. R2a. Supplementary Fig. 11 corresponds to Fig. R3a. Figure R4a has been labeled as Supplementary Fig. 15. The following statement has been added in the revised manuscript (Page 6):

We note that no obvious trace of C₃N₄ has been detected in all the FTIR and Raman spectra.

Comment 3. The vibration peaks in Fig. S9 are very inconspicuous. The IR spectrum of NC needs to be retested.

Response:

We agree with the reviewer's concern and have therefore retested the FTIR spectra. As shown in Fig. R5 (same as Fig. R2a), typical bands at 3243, 2366 and 1641 cm⁻¹ can be attributed to N-H, C-O and H-O, respectively. The FTIR results indicate the presence of C=N-H species, which is consistent with XPS results. We have also amended the corresponding text in the revised manuscript (Page 6):

Fourier-transform infrared (FTIR) spectroscopy further indicates the formation of N-H species in the π -conjugated network of graphitic carbon (Supplementary Fig. 9).

Figure R5. FTIR spectra of NC and Cu₁/NC. The band at 3243 cm⁻¹ corresponds to N–H stretching mode.

Comment 4. It is confusing how *CO(NOOH) converts to *CON in Fig. 6

Response:

We thank the reviewer for this comment. The *CO(NOOH) intermediate will undergo three electrochemical reduction steps to convert into *CON. The first reduction step involves a H atom attached to the –OH on *CO(NOOH), leading to the release of a H₂O molecule and the formation of *CO(NO). The second reduction step involves a H atom attached to the O atom that is bonded with N. Then, the resultant *CO(NOH) intermediate undergoes the third reduction step, with a H atom attached to the –OH and a H₂O molecule subsequently pulled off. Finally, the *CON intermediate (both C and N atoms are bonded with the catalyst) is obtained. We would like to mention that the structures of all the relevant intermediates in the whole reaction path of the co-reduction of NO₃⁻ and CO₂ have been provided in **Supplementary Fig. 42**. In **Fig. R6** we show the corresponding intermediates from *CO(NOOH) to *CON. We have added more detailed description in the revised manuscript on this reaction path (Page 18):

The *CO(NOOH) intermediate undergoes three sequential electrochemical reduction steps and releases two water molecules to form *CON (10), at which the second C–N coupling takes place utilizing another NO₂H nearby.

Figure R6. The optimized atomic structures of intermediates *CO(NOOH), *CO(NO), *CO(NOH) and *CON during urea formation on NC.

Comment 5. As the authors pointed out in the electrocatalytic performance for urea synthesis, “Due to the much smaller number of required electrons to convert NO_3^- to NO_2^- as compared to urea formation, the FE of NO_2^- product is still limited.” Wouldn't the NO_2^- product be easier to produce with less number of electrons required for conversion? Why is it restricted?

Response:

We are grateful to the reviewer for pointing out this statement that may be misleading to readers. In our manuscript, the sentence before this one ends with “a relatively high proportion of NO_3^- converted into NO_2^- ”. In the sentence referred to by the reviewer, we actually want to say that even though a high proportion of NO_3^- is converted into NO_2^- , the FE of NO_2^- is still limited because only two electrons are transferred. This is according to the following equations:

$$\text{N-selectivity: } \text{N-selectivity} = \frac{M(N)}{M_{\text{total}}(N)} * 100\% \quad (1)$$

$$\text{Faradaic efficiency: } \text{FE}(\%) = \frac{n * F * C * V}{M * Q} * 100\% \quad (2)$$

The numbers of transferred electrons (n) to generate NO_2^- and urea are 2 and 16, respectively. Therefore, although N-selectivity for NO_2^- is relatively high on NC (**Fig. 3a** in the manuscript), the FE of NO_2^- is much lower than that of urea. We agree with the reviewer that NO_2^- product is easier to produce with less number of electrons required for conversion, while we want to clarify here the high FE of urea on NC, which makes the statement rather confusing. In the revised manuscript (Page 9), this sentence has been corrected to avoid the misleading implications:

According to the molar yields of N-containing products, the N-selectivity was obtained, showing a relatively high proportion of NO_3^- converted into NO_2^- . **In spite of this, the FE of NO_2^- product is rather limited as compared with that of urea, given the much smaller number of transferred electrons for conversion into NO_2^- than urea.**

Comment 6. As the authors indicated, “The sustained electrocatalytic activity not only demonstrates the long-term stability of NC, but also indicates that the nitrogen source and the carbon source

originate from NO_3^- and CO_2 rather than from the pyridinic/pyrrolic N and carbon atoms in the catalyst.” Probing the nitrogen source and the carbon source with isotopic labeling or other experiments is more credible.

Response:

We would like to thank the reviewer for this kind suggestion, while actually we have already conducted the isotopic labelling experiments in the present work. As displayed in **Fig. R7** (**Fig. 5d** in the manuscript), ^{15}N -isotope and ^{13}C -isotope labelled experiments were performed under an applied potential of -0.5 V versus RHE with $^{15}\text{NO}_3^-$ and $^{13}\text{CO}_2$ as the electrolyte and feeding gas. The Raman peak of $-\text{NH}_2$ band shows a red shift in $^{15}\text{NO}_3^-$ isotope substitution experiments, and the peaks of $^{12}\text{C}-^{15}\text{N}$, $^{13}\text{C}-^{14}\text{N}$, $^{13}\text{C}-^{15}\text{N}$ are evidently shifted to lower values as compared to $^{12}\text{C}-^{14}\text{N}$. These results verify that both nitrogen and carbon sources of urea originate from the employed feedstocks rather than from the functional groups on NC catalyst. The above discussion has been provided in the revised manuscript (Page 16).

Figure R7. Comparison of the *in situ* Raman spectra under isotope-labelled NO_3^- and CO_2 at -0.5 V versus RHE.

Another support comes from the stability test. In the revision, we have extended the test to a total of 15 runs for the co-reduction reaction of NO_3^- and CO_2 . As shown in **Fig. R8** (**Supplementary Fig. 27** in SI), the electrochemical performance is well sustained, suggesting that the C and N species in the electrocatalyst itself will hardly participate as reactants.

Figure R8. Stability test of NC at -0.5 V versus RHE for 15 cycles.

Comment 7. The peaks of Cu^+ and Cu^0 are similar and difficult to distinguish by Cu 2p XPS spectrum. Please provide Cu LMM spectra to further verify the presence of Cu^+ .

Response:

We are grateful for this valuable comment. The Cu LMM spectra for Cu_1/NC has now been provided to verify the presence of Cu^+ . As shown in **Fig. R9**, the peak at 570 eV can be assigned to Cu^+ , while the peak of metallic Cu is hardly distinguishable. This figure has been added as **Supplementary Fig. 16b** in the revised Supplementary Information.

Figure R9. Cu LMM spectrum of Cu_1/NC .

Response to Reviewer #2

Reviewer 2: Authors investigated a nitrogen-doped carbon catalyst which has been employed to sequentially reduce NO_3^- and CO_2 with a Faradaic efficiency of 62% and urea yield rate of $596.1 \mu\text{g mg}^{-1} \text{h}^{-1}$. Further authors proposed that reversible hydrogenation on the nitrogen functional groups that mitigates the overwhelming reduction of reactants, minimizing the formation of side product and highlighted the sequential reduction of nitrate and CO_2 . This is an interesting “concept”. However, there are some critical questions or suggestions for this manuscript from my aspect, as follows.

Response:

We are grateful to the reviewer for the overall positive feedback on our work and for the valuable suggestions and comments. In response to the reviewer’s concerns regarding the scientific mechanism and certain conclusions, we have addressed these issues by incorporating additional experiments and further discussions in the revised manuscript. We have thoroughly considered the reviewer’s comments, and we believe that the revisions have significantly improved the quality of our research. Please find below our point-by-point response, and we hope that the reviewer will find our revisions satisfactory.

Comment 1. In the study, from figure S12 and S13 in supporting information, the surface area on NC is much larger than Cu-NC (close to 5 times higher on NC than Cu-NC), while the urea yield rate on NC is less than three times of that on Cu-NC. This indicates that the intrinsic catalytic activity on Cu-NC for urea production is still higher than the NC catalysts even though the Cu-NC has poor selectivity.

Response:

We would like to thank the reviewer for this valuable comment, and we fully agree with the reviewer that surface area is critical to the catalytic activity. We apologize for not taking into account this issue in our manuscript. According to this comment, we have reexamined our BET analysis and found out that in our previous test, we mistakenly used a short degassing time for the Cu_1/NC sample. A degassing time of 2 h was used in the previous test, with an outcome of $\sim 189.15 \text{ m}^2 \text{ g}^{-1}$ for surface area. But this value increases to $\sim 798.46 \text{ m}^2 \text{ g}^{-1}$ after we extended the degassing time to 6 h (**Fig. R10**). Apparently, with longer degassing time, more porous channel and area will be detected, leading to more accurate results. We note that the surface area of NC sample is $\sim 879.59 \text{ m}^2 \text{ g}^{-1}$ (degassing time is 6 h), very close to that of Cu_1/NC . This result is in line with our assumption that both samples are similar in structure due to the nearly identical synthesis procedure. A difference likely exists between the $\text{C}=\text{N}-\text{H}$ and $\text{N}-\text{Cu}$ species, while in other aspects, both samples would bear strong similarity to each other.

The original **Supplementary Fig. 13** has now been replaced by **Fig. R10**, and we have provided the following statement in the revised manuscript (Page 6):

Related to this feature is a high specific surface area of $879.59 \text{ m}^2 \text{ g}^{-1}$ (for NC sample) as estimated from the Brunauer-Emmett-Teller (BET) plots, and a similar value holds for Cu_1/NC (Supplementary Fig. 12-13).

Figure R10. N₂ adsorption and desorption isotherm for Cu₁/NC. The BET surface area of Cu₁/NC is ~798.46 m² g⁻¹. Inset: pore size distribution for Cu₁/NC.

Comment 2. In page 10, is that solid to claim: “Given that NC and Cu₁/NC were synthesized using the same method with only difference in whether Cu ions were incorporated, both catalysts may share nearly identical functional groups except for the C=N–H and N–Cu species.” by the Figure 2(b) and (e)? In Fig. 2(e) original gray line in the XPS indicates its structure is more complicated. Is that possible proves this by removing the Cu species from the Cu-NC catalysts and then conduct the co-reduction? Besides, during operating conditions (at applied potential at -0.3 to -0.7 V) is it possible that the Cu will leach out and form C=N–H bond?

Response:

We would like to thank the reviewer for this comment. The statement “Given that NC ...” is in fact a speculation based on our synthesis scheme and the XPS results. We agree with the reviewer that the XPS spectrum (**Fig. 2e** in the manuscript) for Cu₁/NC is somewhat irregular, probably due to the complicated structures accompanying the incorporation of Cu (there could be a diverse local environment for Cu atom, apart from the Cu-N₄ moiety shown in **Fig. 2f**). Some functional groups that we could hardly decipher at present may emerge on Cu₁/NC. Nevertheless, the overall profile of this XPS spectrum is still similar to that of the NC sample, except for the observed sharp peak at 399.5 eV. This could be ascribed to the replacement of C=N–H species in NC to N–Cu in Cu₁/NC, the process of which has been demonstrated feasible in the synthesis of similar single atom catalysts (Hai et al. *Nat. Nanotechnol.*, 2022, 17, 174-181). Therefore, we believe that the C=N–H and N–Cu species are likely the main reason behind the difference in catalytic performance, and this can be justified by the proposed sequential co-reduction mechanism according to the *in-situ* spectroscopic measurements and DFT calculations. Moreover, the inferior activity of In₁/NC and Fe₁/NC in synthesizing urea (**Supplementary Fig. 25** in SI) can also substantiate our conclusions.

The speculation regarding the role of C=N–H and N–Cu species is central to our elucidation of the co-reduction reaction process. Yet, we understand that the evidence is not so direct in our work to justify the statement “share nearly identical functional groups except for the C=N–H and N–Cu species”. Therefore, we have corrected this sentence in the revised manuscript (Page 10) to make it more objective and valid:

Given the nearly identical synthesis procedure for both NC and Cu₁/NC, their major difference likely stems from the C=N–H and N–Cu species based on whether Cu ions were incorporated.

The removal of Cu species in Cu₁/NC is feasible upon acid treatment, but thorough removal is rather difficult and the changes of the remaining functional groups are generally inevitable. We have tried using aqua regia to leach the Cu-based species, and found that Cu can only be partially removed according to the ICP data. An incomplete removal of Cu species in Cu₁/NC along with altered functional groups could barely allow a good comparison with the NC sample. In this sense, only limited information can be derived from the co-reduction performance of the Cu₁/NC sample with Cu removed.

To examine whether Cu will leach out to form C=N–H bond in Cu₁/NC, we have evaluated the Cu content before and after the co-reduction reaction according to the XPS spectra (Fig. R11). The Cu content is found to be constant within the limits of experimental error, indicating the negligible leaching of Cu during operation conditions. Figure R11 has been added as Supplementary Fig. 30 in the revised Supplementary Information.

Figure R11. XPS spectra of Cu₁/NC before and after the co-reduction reaction at –0.5 V versus RHE. These XPS data were collected on carbon paper.

Comment 3. "The reaction preference is inherently controlled by the number of N–H bonds and is switchable between favouring NtrRR process or CO₂RR process" is doubtful. The preference of Nitrate reduction over CO₂ reduction during Co-reduction of CO₂ and Nitrate is intrinsic due to a higher reduction potential of Nitrate where it happens at even positive potential while the CO formation from CO₂ requires a more negative potential. The mismatch between these two reduction potentials leads to this low FE of urea. For NC catalyst, it has a higher overpotential to reduce the Nitrate to NH₃, this might bring the reduction potential difference between CO₂ and NO₃⁻ smaller, leaving the possibility of C-N coupling instead of all NO₃⁻ becomes NH₃ which seems happening on Cu-N-C. This can be seen in Fig. 4(a) and (b). It seems NC is an extremely poor nitrate reduction catalyst where nitrate reduction starts at potential < -0.4 V and at this potential CO forms from CO₂. All this makes "proton-involved dynamic catalyst evolution" questionable. And further the results on Fig. 5 are not sufficient to prove this "proton-involved dynamic catalyst evolution" for higher urea selectivity.

Response:

We would like to thank the reviewer for this pertinent comment. Actually, similar discussions have already been provided in our manuscript with respect to the comparison between the activities of NtrRR and CO₂RR (in the **Control experiments for mechanistic rationalization** Section, at Page 11-14). For Cu₁/NC, we emphasize that **“With the balance between NtrRR and CO₂RR struck during co-reduction of NO₃⁻ and CO₂, there would be limited opportunity for urea formation at a single catalytic centre if the activity of this site remains unchanged in the reactions”**. This corresponds to the fierce competition scenario that leads to the overwhelming formation of side products, as illustrated in **Fig. 1a** in the manuscript. We agree with the reviewer that the large mismatch between the onset potentials of NtrRR and CO₂RR is the main reason behind the low FE of urea on Cu₁/NC, and accordingly, a reduced mismatch in reduction potential will definitely benefit C–N coupling and urea formation. This is supported by **Fig. 4c**, in which we display the ratio between NtrRR and CO₂RR current densities in the control experiments, and we conclude that **“NtrRR on NC occupies a less predominating position than that on Cu₁/NC, which could offer more chance for C–N coupling during the co-reduction reaction”**. Overall, we fully subscribe to the idea that the suppressed activity of NtrRR on NC is very important to its high activity for synthesizing urea.

However, the above discussions are from a *macroscopic* perspective, which describe the general trend in the competition between urea formation and side products formation. But this trend cannot fully explain the underlying reason behind the promising urea selectivity on NC. It is worth noting that even on NC, the activity of NtrRR is still much higher than CO₂RR, with the onset potential of NtrRR more positive than CO₂RR (**Fig. R12**, also provided in SI). The current densities for NtrRR and CO₂RR on NC at -0.4 V are 0.30 and 0.015 mA cm⁻², respectively. In this sense, we cannot agree with the reviewer’s claim that *“NC is an extremely poor nitrate reduction catalyst”*. We believe that there is another factor determining the directional synthesis of urea on NC.

Figure R12. Linear sweep voltammetry (LSV) curves of individual NtrRR and individual CO₂RR on NC. The onset potentials for NtrRR and CO₂RR are estimated to be -0.32 and -0.50 V, respectively.

In order to obtain a *microscopic* picture for this directional urea synthesis process, we rely on three independent sets of data: conversion rates of NO₃⁻ and CO₂ (**Fig. 4d and e** in the manuscript), ATR-

SEIRAS characterization (**Fig. 5**) and DFT calculations (**Fig. 6**). The conversion rates of NO_3^- and CO_2 imply that NtrRR precedes CO_2RR in the reaction pathway of urea formation, and the activity of CO_2RR is to some extent promoted by the NtrRR process. The ATR-SEIRAS characterization results demonstrate that the protons on existing $\text{C}=\text{N}-\text{H}$ species at NC catalyst are diminishing at an initial stage of NtrRR. We have performed similar characterization on Cu_1/NC as the reviewer suggested (*Comment 6*), and confirmed that there is no obvious sign of $\text{N}-\text{H}$ depletion, probably responsible for the limited formation of $\text{C}-\text{N}$ bonds (**Fig. R13** and **R14**). The DFT calculations unveil that the catalytic centre on NC is dynamic and reversible in the urea formation process, while the catalytic centre on Cu_1/NC is continuously occupied by the NtrRR reaction intermediates, thus reducing the opportunity for CO_2RR and $\text{C}-\text{N}$ coupling.

These results jointly suggest that the high urea selectivity on NC is closely related with the proton-involved dynamic catalyst evolution. We believe that this causal relationship also exists in some of the recently proposed catalysts for urea synthesis. More importantly, from the mechanistic rationalization in this work, we can foresee the possibility of time-staggering of dual reactions to enable $\text{C}-\text{N}$ coupling for the synthesis of not only urea but also other valuable chemicals. This idea has not been put forward before to the best of our knowledge.

Figure R13. ATR-SEIRAS spectra for Cu_1/NC under different applied potentials during co-reduction of CO_2 and NO_3^- .

Figure R14. ATR-SEIRAS spectra for Cu₁/NC in the range under relatively low applied potentials during individual NtrRR.

Comment 4. Why some catalysts with high urea production yield in Table S1 are not included in Fig.3 (d)?

Response:

We would like to thank the reviewer for pointing out this inconsistency, and we apologize for the insufficiently careful checking of our presentations. Yet, due to space limitations in Fig. 3d, we cannot include all the items in Table S1. Because some studies employed NO₂⁻/NO/N₂ as the N source, which makes it not quite suitable to directly compare with our work, we have discarded them in Fig. 3d (still reserved in Table S1) in the revision. Aside from this, we have discarded those either using a different metric for yield rate or operating at exceedingly negative potentials in the co-reduction reaction. The updated figure is now as follow:

Figure R15. Comparison of urea formation rate and FE between NC and other catalysts reported in the literatures.

Comment 5. In stability test, is “5 successive runs” sufficient? Have you run longer cycles? Comparing Fig.3(b) and Fig. S27, the XPS before and after stability test indeed has some difference. The scattering circle markers are so different, how confident we are can still claim “showing that the structure, morphology and composition of NC catalyst remain largely intact.”?

Response:

We would like to thank the reviewer for this comment. We have followed the reviewer’s suggestion and extended the stability test to a total of 15 runs. As shown in **Fig. R16**, the current density is well maintained for up to 12 runs. The sudden reduction in current density at the 13th cycle is due to the detachment of catalyst materials from the carbon paper upon the continuous flow of CO₂ bubbles. This figure is added as **Supplementary Fig. 27** in SI.

Figure R16. Stability test of NC at -0.5 V versus RHE for 15 cycles. Performance was well maintained for the first 12 cycles. In the 13th cycle, some of the active materials were blown off by CO₂ gas, leading to sudden reduction in current density.

We agree with the reviewer that there is obvious difference between the XPS spectra before and after the stability test. Actually, while the XPS before stability test used the synthesized NC powder, the XPS after test used the materials from the working electrode, which includes carbon paper. The carbon paper will interfere the XPS characterization, leading to low accuracy. We have reconducted the experiments but got similar results. Moreover, some reaction intermediates will remain on the catalyst, contributing to an increased amount of N species. Therefore, we have amended our statement in the revised manuscript (Page 11), as well as the figure caption of the corresponding XPS spectrum (**Supplementary Fig. 29** in SI, here **Fig. R17**).

TEM measurements were further performed after the test (**Supplementary Fig. 28**), showing that the morphology of NC catalyst remains largely intact.

Figure R17. N 1s XPS spectrum of NC after the stability test. Carbon paper at the working electrode will affect the accuracy of measurement to some extent.

Comment 6. Do the authors do the similar analysis (Fig. 5(b)) on Cu-NC? Do authors think that is that also possible that the formation of N-H bond under reduction conditions on Cu-NC?

Response:

We would like to thank the reviewer for this suggestion. Operando ATR-SEIRAS has been conducted on Cu₁/NC for both co-reduction reaction and individual NtrRR, and the results are shown in **Fig. R13 and R14** (see Response to *Comment 3*). As compared to NC (**Fig. 5** in the manuscript), Cu₁/NC exhibits high selectivity to NH₃ formation, as indicated from the marked increase in the height of the peaks corresponding to -NH₂, along with the insignificant changes in the peaks of C-N and C=O (**Fig. R13**). Notably, under low applied potentials during individual NtrRR, the peak corresponding to N-H bonds shows a relatively slight evolution (**Fig. R14**), meaning that only a small amount of N-H bonds are formed/cleaved. We believe that this is inherent to the high stability of the metal-N₄ moiety against hydrogenation (Wang et al. *Nat. Rev. Chem.*, 2018, 2, 65-81; Beniya et al. *Nat. Catal.*, 2019, 2, 590-602). Although we cannot entirely rule out the possibility of forming N-H bonds under reduction conditions on Cu₁/NC, its influence is likely overshadowed by the strong interaction between the NtrRR intermediates and the Cu-N₄ moiety (**Supplementary Fig. 44-45** in SI). With strong binding affinity, the NtrRR intermediates tend to continuously occupy the metal centre, which makes them highly prone to accept the proton on the N atom via a proton-coupled electron transfer step. This is different from the case of NC catalyst, where the NtrRR intermediates such as *NOOH and *NO are physically rather than chemically adsorbed to the catalyst, leaving ample opportunity for CO₂RR and C-N coupling.

Figure R13 and R14 have been added into SI as **Supplementary Fig. 36 and 37**. In the revised manuscript, we have made the following statement (Page 16):

We note that this N-H depletion feature is barely discernible on Cu₁/NC in operando ATR-SEIRAS (Supplementary Fig. 36 and 37).

Comment 7. In the computational details, “The Gibbs free energy in every elementary step was calculated by the computational hydrogen electrode (CHE) model.” Please add appropriate citation for this model.

Response:

We thank the reviewer for this comment, and we have added the citation for this model in the revised Supplementary Information.

[4] Nørskov, J. K. et al. Origin of the overpotential for oxygen reduction at a fuel-cell cathode. *J. Phys. Chem. B* **108**, 17886-17892 (2004).

Comment 8. Sentence in page 5: “..... which is much superior to the previously reported catalysts.” Seems only faradaic efficiency is better while yield rate still can be comparable to some reported literatures (Table S1) So, authors need rewrite it a bit.

Response:

We would like to thank the reviewer and have corrected the statement according to this comment: (Page 5)

Benefiting from these features, a urea yield rate of 596.1 $\mu\text{g mg}^{-1} \text{h}^{-1}$ with a Faradaic efficiency (FE) of 62% was achieved on NC at -0.5 V versus reversible hydrogen electrode (RHE), which is superior to most of the previously reported catalysts.

(Page 10)

Notably, the NC catalyst enables urea electrosynthesis at a maximum yield rate of 596.1 $\mu\text{g mg}^{-1} \text{h}^{-1}$ with a promising FE of 62% under -0.5 versus RHE, which is superior to most of the recently reported catalysts working at similar potentials (Fig. 3d).

REVIEWER COMMENTS

Reviewer #1 (Remarks to the Author):

The electrocatalytic urea synthesis under ambient conditions serves as a promising alternative to the traditional energy-intensive protocol. Li et al. proposed selective urea electrosynthesis on a single-atom copper electrocatalyst. However, there are fatal defects in the product determination and mechanism research of this manuscript. I'd suggest the rejection of this work, follows are my concerns.

1. The diacetyl monoxime method cannot be used to determine the concentration of urea in the presence of nitrate reduction reaction, since the NO_2^- can react with the color generation reagent to produce a false positive result (Chemical Engineering Journal 2023, 453, 139836; J. Am. Chem. Soc. 2022, 144, 11530), therefore the reported data are not credible. The full UV-vis adsorption curves for product determination must be provided, not just the adsorption intensity values.
2. In the DFT part (Figure 6a), the desorption of $^*\text{CO}$ and the 1st C-N coupling deliver energy uphill of 0.65 eV and 0.53 eV respectively, meanwhile, the desorption of $^*\text{CO}$ directly produces gaseous CO whereas non-electrochemical and electrochemical steps must be needed for urea production, so I don't think which is more favorable to the urea formation. Notably, the CO_2 cannot be chemically adsorbed on the catalyst of NC in Figure 6a, implying that the CO_2 reduction reaction seems impossible without the participation of nitrate which is inconsistent with the electrochemical test results.
3. Only thermodynamic parameters are discussed in this work, however, the C-N coupling may be a kinetics-dominated process. The mechanism proposed by the authors possibly owns a large energy barrier, whether it is really suitable for C-N coupling for urea synthesis. So the clear and detailed kinetic studies need to be supplemented.
4. The authors simply proposed the urea synthesis originating from the C=N-H species, which is not concreted and convincing. In order to further verify this conclusion, the necessary electrochemical activities and microstructure characterization contrast of catalyst with different functional groups must be provided. So the introduction of another site (copper) is not reasonable enough in this work.

Reviewer #2 (Remarks to the Author):

Summary:

Thank authors for replying our comments. This is great helpful to clear my doubts. I would suggest editor to accept it to be published.

Response to Reviewer #1

Reviewer 1: The electrocatalytic urea synthesis under ambient conditions serves as a promising alternative to the traditional energy-intensive protocol. Li et al. proposed selective urea electrosynthesis on a single-atom copper electrocatalyst. However, there are fatal defects in the product determination and mechanism research of this manuscript. I'd suggest the rejection of this work, follows are my concerns.

Response:

We would like to thank the reviewer for the pertinent comments on our manuscript. In response to the reviewer's concerns regarding the product determination and the underlying mechanism, we have provided additional experimental and computational evidences to address these issues. We have thoroughly considered the reviewer's comments, and we believe that the revisions have significantly improved the quality of our research. Please find below our point-by-point response, and we hope that the reviewer will find our revisions satisfactory.

Comment 1. The diacetyl monoxime method cannot be used to determine the concentration of urea in the presence of nitrate reduction reaction, since the NO_2^- can react with the color generation reagent to produce a false positive result (Chemical Engineering Journal 2023, 453, 139836; J. Am. Chem. Soc. 2022, 144, 11530), therefore the reported data are not credible. The full UV-vis adsorption curves for product determination must be provided, not just the adsorption intensity values.

Response:

We would like to thank the reviewer for this valuable comment. We agree with the reviewer that the determination of urea concentration is an important issue in this work. Actually, we have spent over one year searching for the appropriate quantification method, and we found that for the electrolyte system employed in this work, the diacetyl monoxime thiosemicarbazide (DAMO-TSC) method is more credible than other methods such as urease method, high performance liquid chromatography-mass spectrometry (HPLC-MS), and nuclear magnetic resonance (NMR) spectroscopy. In the following we compare the efficacy of these four methods for urea determination.

(1) The urease method. The urease method determines the concentration of urea by calculating the concentration difference of ammonia before and after the urease decomposition in the electrolyte. It is very sensitive to pH, temperature and the dissolved heavy metal ions. When 0.1 M KHCO_3 and 0.1 M KNO_3 are used as the background solution in the urease method, we could not get a good linear relationship with different urea concentrations. This is probably due to the influence of NO_2^- and NH_4^+ byproducts, as demonstrated in **Fig. R1**. Similar conclusions were drawn in previous studies (Huang et al. *ACS Energy Lett.*, 2022, 7, 284-291; Li et al. *Small Methods*, 2022, 6, 2200561). Only when the concentration of both influential species is relatively low (on the scale of 0.01 M) can the urease method be precise enough to detect urea accurately (Feng et al. *Nano Lett.*, 2020, 20, 8282-8289; Cao et al. *J. Colloid Interface Sci.*, 2020, 577 109-114.). However, as we cannot get rid of the influence of

NO_2^- and NH_4^+ in our experiments, and the pH of the electrolytes could vary depending on the byproducts, the urease method is definitely not a good choice for urea detection in this work.

Figure R1. (a) Concentrations of urea detected by urease method after adding 0.1 M interferences. The initial urea concentration is set to be 2 ppm. (c) The impact of the concentration of NH_4^+ on the detected concentration of urea via the urease method. Figures are extracted from *Chem. Eng. J.*, 2023, 453, 139836. [Reprint from Yanmei Huang, Yuting Wang, Yang Liu, Aijing Ma, Jianzhou Gui, Chaoxin Zhang, Yifu Yu, Bin Zhang, Unveiling the quantification minefield in electrocatalytic urea synthesis, *Chemical Engineering Journal*, Volume 453, Part 1, 2023, 139836, ISSN 1385-8947, <https://doi.org/10.1016/j.cej.2022.139836>. Copyright (2022), with permission from Elsevier.]

(2) The HPLC-MS method. Most of previous studies were carried out on a C18 column in HPLC for urea determination (Meng et al. *Cell Rep. Phys. Sci.*, 2021, 2, 100378). We have performed a test using a C18 column for various urea concentrations. At concentrations above 10 ppm, the measurements are reliable, with the peak area correlated with the urea concentration, similar to the results of previous studies (Huang et al. *Chem. Eng. J.*, 2023, 453, 139836). However, after the concentration drops to as low as 5 ppm, the peak of urea becomes hardly discernible with the interference from KHCO_3 and KNO_3 in the solution, as shown in **Fig. R2**. To overcome this problem, we tried to use the Hilic column (Agilent) for HPLC-MS measurements, which can potentially alleviate the interference from other ions in the electrolyte. We have compared the results between 0.1 M KHCO_3 + 0.1 M KNO_3 electrolyte and pure water, with 0.5 ppm urea added in the solutions (**Fig. R3**). While the peak of urea does appear, the neutral loss (NL) values are remarkably different in both samples (2.66×10^4 versus 3.36×10^6). This implies that the influence from KHCO_3 and KNO_3 are still non-negligible. Moreover, we further examined the time-dependent variation profiles of the HPLC-MS spectra, and it is surprised to find that there is no variation for the 0.1 M KHCO_3 + 0.1 M KNO_3 electrolyte, in contrast to that of pure water sample (**Fig. R4**). This implies that the peak of urea detected in the 0.1 M KHCO_3 + 0.1 M KNO_3 electrolyte is most likely attributed to the residuals of urea in the column. Overall, from what we observed in the HPLC-MS measurements, we do not think the HPLC-MS method is reliable with respect to low urea concentration in the presence of other ions at high concentrations, at least for the electrolyte system we consider.

Figure R2. The HPLC (C18 column) spectra for 5 ppm (black) and 10 ppm (red) urea in 0.1 M KHCO_3 + 0.1 M KNO_3 electrolyte.

Figure R3. The HPLC-MS (Hilic column) spectra of standard solutions containing 0.5 ppm urea in (a) 0.1 M KHCO_3 + 0.1 M KNO_3 , and (b) pure water.

Figure R4. Time-dependent variation profiles of the HPLC-MS (Hilic column) spectra of standard

solutions containing 0.5 ppm urea in (a) 0.1 M KHCO₃ + 0.1 M KNO₃, and (b) pure water.

(3) The NMR method. The limit of detection via NMR is ~2 ppm according to previous reports (Huang et al. *Chem. Eng. J.*, 2023, 453, 139836). We have performed NMR measurements on a Bruker 500 MHz AVANCE NEO spectrometer equipped with a cryoprobe. The 400 μ L of extracted electrolyte (2 ppm urea + 0.1 M KHCO₃ + 0.1 M KNO₃) was mixed with 200 μ L DMSO-d₆. As displayed in **Fig. R5**, there is no discernible peak corresponding to urea (chemical shift \approx 5.6 ppm) in NMR spectrum. This is probably due to the low signal-to-noise ratio for such a limited concentration of urea. Therefore, the NMR method was not selected for urea determination in this study.

Figure R5. The NMR spectrum of the standard solution containing 2 ppm urea in 0.1 M KHCO₃ + 0.1 M KNO₃ electrolyte.

(4) The DAMO-TSC method. DAMO-TSC method is especially apt to be with cases of low urea concentrations (Li et al. *Small Methods*, 2022, 6, 2200561). We agree with the reviewer that NO₂⁻ can affect the quantification of urea in DAMO-TSC, but we would like to point out that **the influence is trivial when the concentration of NO₂⁻ is relatively low**. We have compared the results of various concentrations of NO₂⁻ (from 0.2 mM to 2 mM) in the solution containing 2 ppm urea. From the UV-vis adsorption curves in **Fig. R6 (Supplementary Fig. 23)**, we can see that the quantification results of urea are within an error of 5 % when the concentration of NO₂⁻ is 0.2 ~ 0.5 mM. Only when the NO₂⁻ concentration is increased to 1 mM can the influence add up to a nontrivial amount, which is in line with the published literature (Wei et al. *J. Am. Chem. Soc.*, 2022, 144, 11530). Notably, in this work, the concentration of NO₂⁻ is measured to be about 0.2 ~ 0.3 mM. Therefore, we believe that the DAMO-TSC method is reliable in determining the concentration of urea in this work.

Figure R6. UV-vis absorption spectra of standard solutions containing 2 ppm urea with various NO_2^- concentrations in DAMO-TSC measurements.

To sum up, the above results fully demonstrate that DAMO-TSC method showcases competitive capability of determining the concentration of urea in this work. We note that all of the methods mentioned above cannot get rid of the influence of other substances in the electrolyte, and the derived urea concentration is always more or less inaccurate. By comparison of these methods from our own experimentation as well as the published literature, we finally ended up with the selection of DAMO-TSC. This method can still provide a somewhat reliable estimation of the urea concentration, which exhibits a reasonable trend that matches well with the operando characterization results and the DFT calculations. We believe that our results are credible, and we have added the full UV-vis adsorption curves in SI (Supplementary Fig. 19-22) according to the reviewer's comment.

Figure R7. Quantification of urea by the diacetyl monoxime method in 0.1 M KHCO_3 and 0.1 M KNO_3 electrolyte. (a) UV-vis absorption spectra. (b) Standard curve of urea concentration.

Figure R8. Quantification of NH_3 by the indophenol blue method in 0.1 M KHCO_3 and 0.1 M KNO_3 electrolyte. (a) UV-vis absorption spectra. (b) Standard curve of NH_3 concentration.

Figure R9. Quantification of NO_2^- by Griess test in 0.1 M KHCO_3 and 0.1 M KNO_3 electrolyte. (a) UV-vis absorption spectra. (b) Standard curve of NO_2^- concentration.

Figure R10. Quantification of N_2H_4 by Watt and Chrisp's method¹⁰ in 0.1 M KHCO_3 and 0.1 M KNO_3 electrolyte. (a) UV-vis absorption spectra. (b) Standard curve of N_2H_4 concentration.

Comment 2. In the DFT part (Figure 6a), the desorption of *CO and the 1st C-N coupling deliver energy uphill of 0.65 eV and 0.53 eV respectively, meanwhile, the desorption of *CO directly produces gaseous CO whereas non-electrochemical and electrochemical steps must be needed for urea production, so I don't think which is more favorable to the urea formation. Notably, the CO₂ cannot be chemically adsorbed on the catalyst of NC in Figure 6a, implying that the CO₂ reduction reaction seems impossible without the participation of nitrate which is inconsistent with the electrochemical test results.

Response:

We would like to thank the reviewer for this insightful comment. We agree with the opinion that the free energy change for the 1st C–N coupling (**7** → **9**) is quite close to that for *CO desorption (**7** → **8**), which seems not compatible with the high selectivity of urea formation on NC catalyst. Since both **7** → **8** and **7** → **9** are non-electrochemical steps, kinetic energy barrier is definitely more reliable than free energy change in evaluating the feasibility of both reaction paths, which has been pointed out by the reviewer in *Comment 3*. In light of the reviewer's suggestion, we have conducted the climbing-image nudged elastic band (CI-NEB) calculations to obtain the transition states in both **7** → **8** and **7** → **9** steps. This method allows for accurate determination of kinetic energy barrier for an elementary step, and it turns out that *CO desorption owns a much larger energy barrier (1.51 eV) than that of C–N coupling (0.72 eV), as shown in **Fig. R11** (**Fig. 6** in the revised manuscript). It is worth noting that in our previous DFT calculations, four water molecules have been introduced to simulate the contribution of hydrogen bonds (see Experimental Procedures part in SI). In the investigation of C–N coupling, this setting was also employed in CI-NEB calculations, and we have carefully searched for the initial and final states to avoid too large variation in the configuration of water molecules, otherwise they may give rise to some artificial perturbation in the energy profile of the reaction. We find that the transition state in C–N coupling is quite similar to the final state *CO(NO₂H), with the main geometric difference lying in the number of hydrogen bonds (**Fig. R12**). This indicates that the self-adjusted hydrogen bond network could to some extent modulate the configuration in the transition state, and facilitate the facile establishment of the C–N bond. On the other hand, during *CO desorption (**Fig. R13**), the most significant change is the number of C–N bonds (here the N atoms belong to the NC catalyst). In the initial state, there are two C–N bonds, along with the C=O double bond in *CO species. In the final state, there is a C≡O triple bond in the CO molecule. Both of these states are rather stable, while the transition state, with one remaining C–N bond, is highly unstable in the sense that it does not satisfy the 8-electron rule. Obviously, there is a radical on the C atom in this configuration, leading to a remarkably high energy barrier as compared to the moderate free energy change between the initial and final states.

We have added **Fig. R12** and **Fig. R13** in SI, and amended the corresponding text in the revised manuscript (Page 18):

Since both steps are kinetics-dominated processes, we here employed climbing-image nudged elastic band (CI-NEB) calculations (Supplementary Fig. 44) to determine their activation barriers. The results show that the barrier for the formation of CO(g) (**8**) is substantially higher than that of *CO(NO₂H) (**9**), well-matching with the inferior selectivity to CO product in the experiments.

Figure R11. Proposed reaction path of the sequential co-reduction reaction on NC. (a) The DFT-calculated Gibbs free energy profile of the co-reduction reaction of NO_3^- and CO_2 at 0 V versus RHE. Elementary steps marked by dashed lines are less favourable than those marked by solid lines. The transition states (TS) of $^*\text{CO}$ desorption ($7 \rightarrow 8$) and C–N coupling ($7 \rightarrow 9$) were obtained by CI-NEB calculations. (b) Structural configurations of the critical reaction intermediates. Intermediates in the blue frame are related to the sequential reduction mechanism leading to urea formation.

Figure R12. CI-NEB calculation for the 1st C-N coupling process on NC. The structural configurations of the initial state (IS), transition state (TS) and final state (FS) are displayed.

Figure R13. CI-NEB calculation for *CO desorption process on NC. The structural configurations of the initial state (IS), transition state (TS) and final state (FS) are displayed.

The reviewer commented that “the CO₂ cannot be chemically adsorbed on the catalyst of NC in Figure 6a, implying that the CO₂ reduction reaction seems impossible without the participation of nitrate”. While it is true that CO₂ can only be physically adsorbed on NC, we could hardly agree with the reviewer that it is impossible for CO₂RR to take place without nitrate. We would rather use the word “unfavourable” instead of “impossible”. Typically, we should not simply rely on the structural configurations when deducing whether a reaction could take place or not. It is worth noting that

*NOOH (**4**) is physically rather than chemically adsorbed to the catalyst, as mentioned in our manuscript. The way NO₂H is adsorbed does not affect its participation in C–N coupling, and even its reduction into NH₃, although the path to NH₃ is not very facile on NC catalyst. To determine whether a reaction path is favourable, we should rely on the free energy change and the activation barrier. But even if we can tell which path is favourable, it is by no means appropriate to say the other reaction paths are impossible. DFT calculations could only provide a qualitative understanding of how the reaction is steered towards the target product, but could not rule out the possibility of forming other products. This is because the calculations are restricted to ideal cases, while the real condition on the catalysts can be rather complicated. We believe that CO₂RR can still take place without nitrate despite the high energy uphill in this path. The control experiments in **Fig. 4** in the manuscript suggest that the current density of individual CO₂RR on NC is less than one tenth that of individual NtrRR. This indicates that CO₂RR is very unfavourable on NC as compared to NtrRR, which is consistent with the implications from DFT free-energy calculations.

***Comment 3.** Only thermodynamic parameters are discussed in this work, however, the C-N coupling may be a kinetics-dominated process. The mechanism proposed by the authors possibly owns a large energy barrier, whether it is really suitable for C-N coupling for urea synthesis. So the clear and detailed kinetic studies need to be supplemented.*

Response:

We agree with the reviewer's concern and have therefore performed the climbing-image nudged elastic band (CI-NEB) calculations to obtain the transition states in both the 1st C–N coupling (**7** → **9**) and *CO desorption (**7** → **8**) steps. The kinetic energy barriers can be accurately predicted from these calculations. All the results have been displayed in the Response to reviewer's *Comment 2*. We note that C–N coupling owns a relatively low energy barrier as compared to *CO desorption, thus providing strong evidence for the proposed reaction path to urea in this work. We are grateful to the reviewer for this valuable suggestion, which has significantly improved the mechanistic understanding regarding the competition between urea and CO formation on NC catalyst.

***Comment 4.** The authors simply proposed the urea synthesis originating from the C=N-H species, which is not concentered and convincing. In order to further verify this conclusion, the necessary electrochemical activities and microstructure characterization contrast of catalyst with different functional groups must be provided. So the introduction of another site (copper) is not reasonable enough in this work.*

Response:

We would like to thank the reviewer for this comment and the thoughtful suggestion to verify the role of C=N–H species in NC. In this work, we try to decipher the catalytic mechanism on NC from the following considerations:

(1) Removal of C=N–H species by the introduction of Cu during synthesis. We note that NC and Cu₁/NC samples were prepared using exactly the same procedure except for whether the Cu precursor

is added. XPS spectra (**Fig. 2** in the manuscript) indicate that the incorporation of Cu leads to the emergence of a sharp peak at 399.5 eV, which could be ascribed to the replacement of C=N–H species in NC catalyst to N–Cu in Cu₁/NC. This process has been demonstrated feasible in the synthesis of single atom catalysts in previous studies (Hai et al. *Nat. Nanotechnol.*, 2022, 17, 174-181). Moreover, BET analysis suggests that the surface areas of both samples are nearly the same (**Supplementary Fig. 12-13** in SI). We have also examined whether Cu will leach out to form C=N–H bond in Cu₁/NC, and we found that Cu content in Cu₁/NC is constant within the limits of experimental error during operation conditions, thus precluding the possibility of transformation of N–Cu into C=N–H (**Supplementary Fig. 33** in SI). Therefore, we believe that the major difference between the structures of both catalysts stems from the C=N–H and N–Cu species, and a comparison in catalytic performance between both samples could serve as an important justification of the role of C=N–H in urea synthesis.

(2) Mechanistic insights from control experiments of individual NtrRR and individual CO₂RR.

We note that according to the DFT calculations for the energetics of N–H bond formation on NC (**Supplementary Fig. 17**), as well as the previous experimental evidence of Faradaic pseudocapacitance behaviour on nitrogen-doped carbon materials (Tian et al. *Nat. Commun.*, 2020, 11, 1-10), it is safe to say that the C=N–H species could undergo reversible formation and cleavage of the N–H bonds during electrochemical reactions. This is compatible with all the observations in the control experiments of individual NtrRR and individual CO₂RR on the NC sample (**Fig. 4** in the manuscript). These observations include: (a) NtrRR precedes CO₂RR and is hardly affected by CO₂RR intermediates; (b) electrochemical conversion of CO₂ is activated in the presence of NO₃[−]. The above results indicate a consecutive switch of catalytic activity from favouring NtrRR to favouring CO₂RR in the reaction path to urea on NC. It means that a dynamic active centre is likely involved in urea synthesis from the co-reduction of NO₃[−] and CO₂. We believe that such a close match between the dynamic-catalyst-evolution mechanism and the reversible-hydrogenation ability of C=N–H species could serve as another justification of the critical role of C=N–H in NC catalyst.

(3) Operando characterization of the N–H species on NC.

We have performed operando ATR-SEIRAS characterization to justify that the protons on existing C=N–H species at NC catalyst are diminishing at an initial stage of NtrRR (**Fig. 5b** in the manuscript). We have also performed similar characterization on Cu₁/NC and confirmed that there is no obvious sign of N–H depletion (**Supplementary Fig. 40**), which is probably inherent to the high stability of the metal-N₄ moiety against hydrogenation (Wang et al. *Nat. Rev. Chem.*, 2018, 2, 65-81; Beniya et al. *Nat. Catal.*, 2019, 2, 590-602). These operando characterizations, in combination with the results of control experiments, indicate that the proton-involved dynamic catalyst evolution is directly correlated with the C=N–H species instead of other functional groups. That is, the NtrRR process consumes the protons on C=N–H species and produces an alteration in catalytic activity, which creates the opportunity to switch the reaction path to CO₂RR and enable the sequential mechanism of urea synthesis. This serves as the third justification of the critical role of C=N–H for urea synthesis.

(4) DFT calculations of the reaction path to urea.

We have performed DFT calculations (**Fig. 6** in the manuscript) for all the intermediates in the reaction path to urea, under the assumption that C=N–H

and N–Cu species are the critical catalytic centres of NC and Cu₁/NC, respectively. Kinetic energy barrier calculations were also conducted, as the reviewer suggested (*Comments 2 and 3*). All the results are consistent with our experimental findings, and can substantiate the role of C=N–H in biasing the competition between NtrRR and CO₂RR to enable urea synthesis. This serves as the fourth justification.

(5) Comparison between samples synthesized at different pyrolysis temperatures. In the revision, we have supplemented a test on the NC catalyst synthesized at an elevated pyrolysis temperature. In our manuscript, the NC catalyst was prepared at 900 °C (denoted as NC-900 in the following text). Here we prepared another sample with the pyrolysis temperature raised to 1000 °C (NC-1000). As demonstrated from the XPS results in **Fig. R14**, the ratio of C=N–H species drops substantially on NC-1000 when compared with NC-900. In **Fig. R15**, we show that the FE of urea on NC-1000 is considerably lower than that on NC-900. We believe that this could serve as the fifth justification of the critical role of C=N–H for urea synthesis.

Figure R14 and R15 have been added as **Supplementary Fig. 30-31** in SI, and the following statement has been added in the revised manuscript (Page 11):

Additionally, we have synthesized another NC sample at an elevated pyrolysis temperature, which results in a greatly reduced amount of C=N–H species and exhibits a much lower urea FE in the co-reduction reaction (Supplementary Fig. 30-31). This implies that C=N–H plays a pivotal role for urea synthesis on NC.

Overall, we have presented five independent sets of evidence suggesting that C=N–H species play an important role in urea synthesis on NC catalyst. We would like to note that it is beyond our capability to adjust one functional group on NC while not interfering with other functional groups. This makes it impossible to provide the “*electrochemical activities and microstructure characterization contrast of catalyst with different functional groups*” as stated by the reviewer. One example is the difference between NC-900 and NC-1000 as mentioned above. While the amount of C=N–H species has changed, the amounts of other functional groups, such as graphitic N, are also altered considerably. Given that graphitic N is conducive to CO formation in CO₂RR according to previous studies (Zhang et al. *Cell Rep. Phys. Sci.*, 2020, 1, 100145; Li et al. *Appl. Catal. B*, 2021, 298, 120510), it is not possible to say that the difference in catalytic performance is solely contributed by the changes in C=N–H species in NC. Therefore, the comparison between NC-900 and NC-1000 is actually qualitative rather than quantitative. Nevertheless, in this work, all of the experimental and computational results unequivocally establish the relevance of C=N–H species to urea synthesis in the co-reduction reaction of NO₃[−] and CO₂. We believe that our conclusions on C=N–H species are sound. We also speculate that the mechanistic insights gained from this study could be extended to other recently proposed catalysts for urea synthesis, and would hopefully inspire the design of more catalysts with time-staggering of dual reactions to enable C–N coupling in electrochemical reactions.

Figure R14. (a) N 1s XPS spectrum of NC-1000. (b) The ratios of C=N-H, graphitic N, pyridinic N, pyrrolic N and oxidized N in NC-900 and NC-1000.

Figure R15. (a) The FE of urea for NC-900 and NC-1000. The color of the solutions after the co-reduction reaction on (b) NC-900 and (c) NC-1000.

Response to Reviewer #2

Reviewer 2: Thank authors for replying our comments. This is great helpful to clear my doubts. I would suggest editor to accept it to be published.

Response:

We thank the reviewer for the positive comments.

REVIEWERS' COMMENTS

Reviewer #1 (Remarks to the Author):

The Authors addressed all my comments, and the manuscript can be accepted in the present version.

Response to Reviewer #1

Reviewer 1: The Authors addressed all my comments, and the manuscript can be accepted in the present version.

Response:

We thank the reviewer for the positive feedback.